# Revisiting Circulating Extracellular Matrix Fragments as Disease Markers in Myelofibrosis and Related Neoplasms

**DOI:** 10.3390/cancers15174323

**Published:** 2023-08-29

**Authors:** Hans Carl Hasselbalch, Peter Junker, Vibe Skov, Lasse Kjær, Trine A. Knudsen, Morten Kranker Larsen, Morten Orebo Holmström, Mads Hald Andersen, Christina Jensen, Morten A. Karsdal, Nicholas Willumsen

**Affiliations:** 1Department of Hematology, Zealand University Hospital, 4000 Roskilde, Denmark; vihs@regionsjaelland.dk (V.S.); laskj@regionsjaelland.dk (L.K.); trak@regionsjaelland.dk (T.A.K.); mokl@regionsjaelland.dk (M.K.L.); 2Department of Rheumatology, Odense University Hospital, 5000 Odense, Denmark; peter.junker@rsyd.dk; 3National Center for Cancer Immune Therapy, Herlev Hospital, 2730 Herlev, Denmark; morten.orebo.holmstroem@regionh.dk (M.O.H.); mads.hald.andersen@regionh.dk (M.H.A.); 4Nordic Bioscience A/S, 2730 Herlev, Denmark; chj@nordicbio.com (C.J.); mk@nordicbio.com (M.A.K.); nwi@nordicbio.com (N.W.)

**Keywords:** myeloproliferative neoplasms, MPNs, MPN, circulating extracellular matrix (ECM) proteins, serum procollagen III N-terminal propeptide (PIIINP), procollagen type I C-terminal propeptide (PICP), laminin, type IV collagen, hyaluronan, neoepitopes, protein fingerprints

## Abstract

**Simple Summary:**

MPN blood cancers are characterized by elevated blood cell counts. Chronic inflammation drives MPNs, which after 10–30 years, may lead to bone marrow failure due to replacement of bone marrow by connective tissue. Chronic inflammation also impacts other organs causing impaired vision, osteoporosis, fractures, heart failure and decreased lung and kidney function. MPNs are also associated with an increased risk of other cancers. Novel, noninvasive technologies are needed to improve the early detection of fibrosis development and MPN progression and to monitor the efficacy of treatment. Herein, we review current knowledge and focus on novel technologies for the measurement of circulating connective tissue biomarkers in MPNs. Future research directions are provided, suggesting studies on single markers or combinations before and during treatment with old and new agents, both targeting the diseased bone marrow cells and the inflammatory processes that drive the progressive fibrosis, which if left untreated, may ultimately lead to bone marrow failure or leukemic transformation.

**Abstract:**

Philadelphia chromosome-negative chronic myeloproliferative neoplasms (MPNs) arise due to acquired somatic driver mutations in stem cells and develop over 10–30 years from the earliest cancer stages (essential thrombocythemia, polycythemia vera) towards the advanced myelofibrosis stage with bone marrow failure. The *JAK2V617F* mutation is the most prevalent driver mutation. Chronic inflammation is considered to be a major pathogenetic player, both as a trigger of MPN development and as a driver of disease progression. Chronic inflammation in MPNs is characterized by persistent connective tissue remodeling, which leads to organ dysfunction and ultimately, organ failure, due to excessive accumulation of extracellular matrix (ECM). Considering that MPNs are acquired clonal stem cell diseases developing in an inflammatory microenvironment in which the hematopoietic cell populations are progressively replaced by stromal proliferation—“a wound that never heals”—we herein aim to provide a comprehensive review of previous promising research in the field of circulating ECM fragments in the diagnosis, treatment and monitoring of MPNs. We address the rationales and highlight new perspectives for the use of circulating ECM protein fragments as biologically plausible, noninvasive disease markers in the management of MPNs.

## 1. Introduction

The classic chronic Philadelphia-negative myeloproliferative neoplasms (MPNs) encompass essential thrombocythemia (ET), polycythemia vera (PV) and primary myelofibrosis (PMF) [1,2]. They arise due to acquired somatic mutations in the pluripotent stem cell, which subsequently give rise to clonal expansion and clonal evolution over decades (10–30 years) in a biological continuum from the early cancer stage ET to the advanced myelofibrosis stage and ultimately leukemic transformation [3]. The somatic “driver” mutations include *JAK2V617F*, *CALR* and *MPL,* which drive the malignant clone throughout the transitions [3]. Among these mutations, the most prevalent is the *JAK2V617F* mutation, which occurs in virtually all patients with PV (95–98%), and in approximately 50–60% of patients with ET or PMF [2]. Additional non-driver mutations are recorded in > 50% of patients with MPNs and are considered to be implicated in clonal evolution with myelofibrotic and leukemic transformation [4,5,6]. In the majority of *JAK2V617F*-negative ET and PMF patients, *CALR* or *MPL* mutations are detectable, leaving approximately 10% of patients with MPNs carrying none of the three driver mutations, the so-called “triple-negative” MPNs [7,8].

MPNs arise from clonal hematopoiesis of indeterminate potential (CHIP), which is an inevitable consequence of normal aging and is defined as the presence of a clonal mutation in a driver gene, occurring with a variant burden of ≥2% but without any clinical evidence of hematologic cancer [9]. Individuals harboring CHIP have an approximately 10-fold increased risk of developing hematologic cancer, including MPNs [10]. Furthermore, similar to patients with MPNs [11,12,13,14,15,16,17], individuals with CHIP have an increased risk of cardiovascular diseases, including coronary artery disease and stroke [18,19].

In recent years, the concept of chronic inflammation as a trigger and driver of MPNs has been increasingly recognized [20,21,22,23,24,25,26,27,28]. Indeed, it has been proposed that MPNs are “A Human Inflammation Model“ and “A Human Inflammation Model of Cancer Development“ [20,21,22,23]. In this regard, chronic inflammation may also trigger and drive the development of accelerated (premature) atherosclerosis in MPNs, as evidenced by a heavy burden of coronary artery and aortic valve calcifications [16,17] and the well-established increased risk of second cancers in MPNs [20,21,23,29,30,31,32,33,34]. The importance of immune deregulation [35,36] with increasing T-cell exhaustion and tumor immune evasion consequent to defective tumor immune surveillance has been repeatedly underscored as one of the major mechanisms, contributing not only to MPN development and disease progression but also to the development of second cancers and their poorer survival as compared to the same cancers in the background population [30,31,34]. Highly intriguing, other inflammation-mediated comorbidities are also prevalent in MPNs, including much earlier development of drusen and age-related macular degeneration [37,38,39,40,41,42], increased risk of fractures, likely mediated by the chronic inflammatory state [43,44,45,46,47,48,49,50], chronic inflammatory bowel diseases (ulcerative colitis, Crohn’s disease) [51,52,53,54], and chronic kidney disease [55,56,57,58,59,60,61,62,63,64,65]. Accordingly, patients with MPNs are not only at a constantly increased risk of major thrombosis in, e.g., the brain, heart, lungs, legs and abdominal arteries and veins [11,12,13,14,15] but also at an increased risk of being burdened by inflammation-mediated multimorbidities [37,38,39,40,41,42,43,44,45,46,47,48,49,50,51,52,53,54,55,56,57,58,59,60,61,62,63,64,65], including an increased risk of second cancers [20,21,22,23,29,30,31,32,33,34]. Therefore, both prevention, diagnosis and treatment of MPNs require close interdisciplinary collaboration between experts within several disciplines, including hematology, oncology, cardiology, neurology, gastroenterology, nephrology, rheumatology, ophthalmology and molecular biology.

As mentioned above, the MPNs arise from CHIP [9,10,18,19] and evolve over decades from the earliest disease stage ET to the “metastatic” advanced myelofibrosis stage with a multitude of comorbidities [37,38,39,40,41,42,43,44,45,46,47,48,49,50,51,52,53,54,55,56,57,58,59,60,61,62,63,64,65,66,67,68,69]. During this evolution, mutated CD34+ stem cells egress from the “inflamed” bone marrow niches into the systemic blood compartment to seed in the spleen, liver and elsewhere (“metastasis”) [70,71,72,73,74,75,76,77,78,79]. Since bone marrow failure due to chronic inflammation and progressive accumulation of fibrous tissue in the bone marrow is the terminal outcome, there is an urgent unmet need to identify biomarkers to monitor the inflammation-mediated fibrotic disease progression and for the selection of individualized therapies based on knowledge about specific disease pathways as adjuncts to conventional means such as blood cell counts, *JAK2V617F* mutational load over time and bone marrow biopsies.

Previous studies on the potential of circulating extracellular matrix (ECM) fragments released to the circulation consequent to the altered MPN-related connective tissue turnover as markers of current early bone marrow inflammation and late fibrogenesis, respectively, have demonstrated highly pertinent associations between circulating levels of several ECM proteins and disease activity and drug responses in MPNs, particularly regarding the procollagen III-N-terminal propeptide, type IV collagen, laminin and hyaluronan [80,81,82,83,84,85,86,87,88,89,90,91,92,93,94,95,96,97,98]. Within recent years, the research field on ECM and circulating ECM fragments has expanded immensely in several diseases, not least within cardiovascular diseases and cancer [99,100,101,102,103,104,105,106,107,108,109,110,111,112,113,114,115,116,117,118,119,120,121,122,123,124,125,126,127,128,129,130,131,132,133,134,135,136,137,138,139,140,141,142,143,144,145,146,147,148,149,150,151,152,153,154,155,156,157]. Importantly, novel technologies have been introduced that detect neoepitopes, referred to as protein fingerprints. These neoepitopes are exposed on the ECM degradation product, generated by, e.g., matrix metalloproteinase (MMP) activity during altered ECM remodelling in cancer and inflammation-mediated diseases [100,101,102,103,104]. Measuring specific neo-epitopes on ECM fragments rather than total protein can provide hitherto unprecedented information about fibrogenesis and fibrosis, collagen formation, MMP-mediated collagen degradation and collagen cross-linking in particular (Figure 1). These ECM fragments are released into the circulation and can be measured with highly sensitive and specific immunoassays providing a noninvasive biomarker tool [100,101,102,103,104].

Based upon experimental and clinical studies, we herein address the rationales and perspectives for measuring circulating ECM fragments in MPNs, aiming to achieve an integrated neoepitope signature for disease activity in terms of myeloproliferation, inflammation, fibrosis and angiogenesis. Also, it is a goal within reach to provide additional proof of concept on circulating ECM fragments as biomarkers of disease activity in MPNs, thereby contributing to elucidating the role of somatic mutations in blood cells as links between chronic inflammation, vascular diseases, and organ fibrosis in the MPNs.

Before addressing the rationales and perspectives of the assessment of circulating ECM fragments as novel tools to monitor disease activity and response to treatment in MPNs, we will briefly review the normal bone marrow stroma, the connective tissue response to injury, including stroma generation in normal wound healing and tumors, and how the bone marrow stroma is altered in MPNs and its impact upon bone marrow function and progression of MPNs towards bone marrow failure and ultimately leukemic transformation.
Figure 1Circulating type III collagen fragments with specific neo-epitopes can be measured with the neo-epitope technology. Cells such as fibroblasts and reticular cells synthesize type III collagen. The type III pro-collagen is released into the extracellular space with subsequent proteolytic removal of the N-terminal propeptide and the C-terminal propeptide leading to collagen fibril formation and intrapeptide and interpeptide cross-linking with enzymes such as lysyl oxidase (LOX). Immune-mediated proteolysis with, e.g., metalloproteinases (MMPs) of the mature type III collagen results in the release of collagen fragments with specific neo-epitopes into the circulation. The neo-epitope technology is based on monoclonal antibodies, which enables the assessment of these specific neo-epitopes in serum or plasma. The biomarker PRO-C3 measures the C-terminal of the N-terminal propeptide, reflecting the true type III collagen formation, while PIIINP measures an internal epitope on the N-terminal propeptide. C3M measures MMP-degraded type III collagen reflecting fibrolysis, while CTX-3 measures cross-linked and MMP-degraded type III collagen.
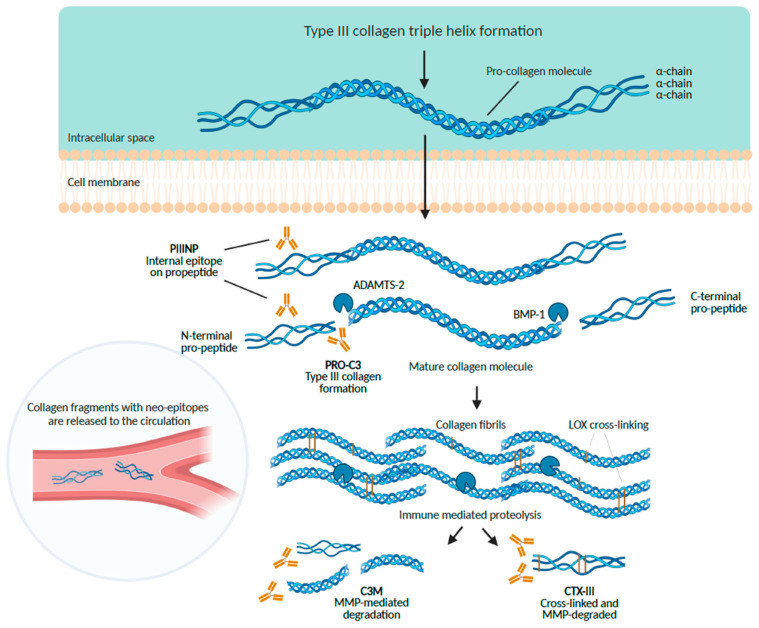


## 2. The Normal Bone Marrow Stroma

The ECM of connective tissue is a highly important constituent of the normal bone marrow compartment and a prerequisite for normal bone marrow function. Connective tissue is composed of cells, fibers and amorphous ground substance, which consists mainly of proteoglycans and hyaluronan with collagens as the principal structural component in the ECM (Figure 2). Fibronectin and laminin are major non-collagenous extracellular glycoproteins that are important for cell adhesion and growth. There are 28 different types of collagen, and each collagen is composed of three polypeptide alfa-chains, which form a triple helical structure. The central helical portion comprises the large majority of the molecule [158,159,160,161]. Fibrillar interstitial collagens comprise types I, II, III, V, XI, XXIV and XXVII. Collagen type I is the most abundant protein in the body and is found in, e.g., the skin, bone, and tendons. Collagen type II is found in hyaline cartilage and the vitreous body. Collagen type III codistributes in normal bone marrow with reticular fibers and also in other tissues such as the skin, blood vessels, synovium, spleen and lymph nodes. Importantly, this collagen type occurs early at sites of injury. Type V, XI, XXIV and XXVII collagen are less abundantly expressed and may serve unique functions by binding to the other fibrillar collagens and supporting fibrillization [161]. Collagen type IV is a major constituent of basement membranes, a specialized ECM compartment in proximity to the epithelia and blood vessel compartments.
Figure 2Stroma of healthy bone marrow, myelofibrosis, healthy solid tissue and tumor fibrosis. The stroma of healthy bone marrow and healthy solid tissues are similar. These tissues are composed of fibroblasts, immune cells, fibers and amorphous ground substance, which consists mainly of proteoglycans, hyaluronan and collagens. The stroma in myelofibrosis and cancer are characterized by excessive inflammation and deposition of collagen fibers such as type I and type III collagen, as well as new blood vessel formation. In myelofibrosis, megakaryocytes are abundant in the bone marrow. Stroma generation in tumors and in bone marrow in myelofibrosis appear to follow an identical sequential pattern with *oedema*, *angiogenesis* and *fibrosis*.
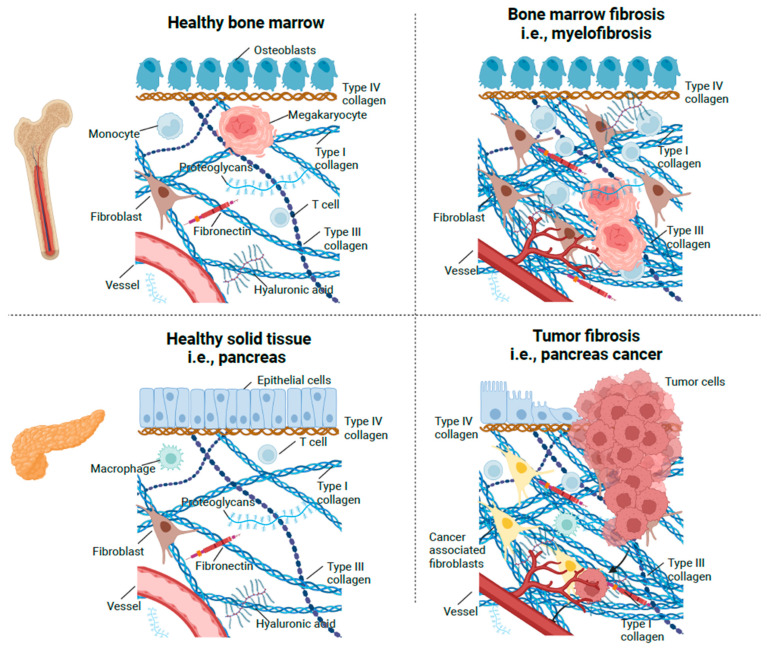


## 3. The Connective Tissue Response to Injury

### 3.1. Stroma Generation in Normal Wound Healing and Tumors

The connective tissue response to injury is characterized by distinctive alterations in the composition of the ECM [161]. This injury–repair process—*oedema*–*angiogenesis*–*fibrosis*—is qualitatively similar in all organs and tissues and independent of the type of injury. Accordingly, the repair processes during normal wound healing and those seen during tumor stroma generation are, in many respects, quite similar [162,163,164]. The differences between normal wound healing and the generation of tumor stroma are principally related to the initiating mechanisms and/or lack of completion mechanisms. Both processes are initially characterized by extravasation of plasma proteins, which in wound healing is triggered by local tissue injury with damage to blood vessels, but in tumor stroma generation mediated by the release of a vascular permeability factor (VPF), which markedly enhances the permeability of blood vessels [165]. While normal vascular permeability is restored within a few days in healing wounds, the secretion of VPF from tumor cells implies persistent hyperpermeability of blood vessels with continuous leakage of plasma proteins, e.g., fibrinogen and fibronectin. In this regard, the importance of increased vascular permeability for stroma formation in tumors, healing wounds, and chronic inflammation has most recently been outlined [164,165], thereby reconciling vascular endothelial growth factor (VEGF) with VPF [165].

In normal wound healing, platelets are extremely important in the generation of a provisional fibrin–fibronectin matrix by releasing clotting factors and factors, which are mitogens and chemoattractants for connective tissue cells such as fibroblasts [166]. By contrast, platelets do not participate in the generation of the fibrin–fibronectin matrix in tumors, whereas tumor cells are able to secrete several factors with virtually identical functions as those released from platelets, including platelet-derived growth factors (PDGFs), VEGF, transforming growth factor beta (TGF-beta), and fibroblast growth factor 2 (FGF-2) [167,168,169].

The inflammatory oedema of wound healing is also generated due to the accumulation of strongly hydrophilic hyaluronan, a high-molecular-weight polysaccharide consisting of alternating glucosamine and glucuronic acid subunits [170,171]. Thereafter, the provisional matrix of fibrin and fibronectin is degraded by invading inflammatory cells, followed by the growth of new capillaries—angiogenesis—with the deposition of basement membrane material, such as collagen type IV and laminin. In this cellular, oedematous and vascular environment, fibroblasts synthesize interstitial collagens with the deposition of type III collagen at the earliest stage and type I collagen in the later phase of fibrogenesis, where the tissue is composed largely of dense collagen and only scattered fibroblasts and blood vessels. The continuous extravasation of fibrinogen and fibronectin from blood vessels adjacent to tumors implies that the healing process is constantly triggered since the signal of this process—the fibrin–fibronectin matrix—undergoes continuous remodeling. Accordingly, tumors have been conceived as wounds that never heal because the stimulus for the healing process persists [162,163,164,165]. This also holds true for MPNs. In this regard, the MPNs constitute a unique model to study ECM metabolism in cancer since they represent the wounding phases at the earliest cancer stage—ET—presenting with “organ dysfunction” only towards the metastatic myelofibrosis stage of “collagenization” with fibrosis and subsequent bone marrow failure.

### 3.2. Bone Marrow Stroma Generation in Primary Myelofibrosis (PMF)

As alluded to above, stroma generation in the bone marrow in PMF and related neoplasms has many similarities with those recorded during wound healing and tumor stroma generation. Thus, in the early hypercellular phase of PMF, the bone marrow stroma shows signs of inflammation with oedema and infiltration with lymphocytes, plasma cells and mast cells. Furthermore, bone marrow stroma is characterized by excess capillary formation [172]. Myeloproliferation is characterized by a predominance of large dysplastic megakaryocytes, often located in clusters close to the increased network of fine reticular fibers, which are composed largely of type III collagen (Figure 2) [87,95].

As MPN diseases progress, hematopoietic tissue, except for megakaryocytes, is gradually replaced by dense Van-Giesson-positive collagen fibers (mainly type I collagen) with or without osteomyelosclerosis. However, even at this stage of the disease, there is also an increased deposition of type III collagen fibers [95], which conceivably reflects a continuous stimulus of repair processes. As compared to the early stage of bone marrow fibrosis, the growth of capillaries is even more intense in the advanced stage [173,174,175,176,177]. In particular, osteomyelosclerosis is accompanied by a marked increase in the number of capillaries, together with the development of continuous sheets of basement membrane material beneath the endothelial cells [95,172,173].

Despite stroma generation in wound healing, tumors and bone marrow in myelofibrosis appear to exhibit an identical sequential pattern with oedema, angiogenesis and fibrosis, the stroma changes in PMF more closely resemble those seen during tumor stroma generation, where angiogenesis persists. As previously mentioned, the stimulus for capillary growth in PMF is considered to be very similar to that observed in tumor stroma generation, implying that the increased vascular permeability with extravasation of plasma proteins accounts for the continuous deposition of a fibrin–fibronectin containing matrix, which evokes capillary growth [164,165]. The increased vascular permeability could be due to altered basement membrane architecture as well as to the liberation of histamin from mast cells, which are always increased in number in PMF [178]. The findings of fibrin deposits [178] and large amounts of fibronectin, mainly having a perivascular distribution, in bone marrow in PMF [179,180], may support the concept of a continuous generation of a fibrin–fibronectin matrix in PMF. Furthermore, angiogenesis and endothelial proliferation [180] are elicited by the release of angiogenic growth factors from rapidly proliferating or necrotic megakaryocytes [181,182]. Indeed, the clustering of megakaryocytes close to sinusoids and their intrasinusoidal localization [183,184] support the concept of regulatory interactions between megakaryocytes and endothelial cells [183,184]. In this regard, megakaryocytes are key regulators of the hematopoietic stem cell niche [184], which is also markedly influenced and regulated by inflammation [185]. Considering the role of chronic inflammation for disease progression in MPNs, sustained chronic inflammation in the stem cell niche may have detrimental effects, leading to hematopoietic stem cell damage and ultimately promoting bone marrow fibrosis and leukemic transformation [186,187]. Importantly, in a murine model of MPN, *JAK2V617F* mutant megakaryocytes not only contributed to hematopoietic stem cell/progenitor cell expansion [188,189] but also to hematopoietic aging [190].

## 4. Histological Studies on Bone Marrow Stroma in MPNs

Histologically, two stages of myelofibrosis have broadly been distinguished, “reticulin fibrosis” and “collagen fibrosis”, with the deposition of silver stainable (=reticulin) fibers in the early hypercellular disease phase and the accumulation of a mature collagen type I rich matrix in the advanced disease phase. However, the silver impregnation technique is not specific to any particular matrix protein. Thus, although reticulin fibers are composed of type III procollagen (pN-collagen) and type III collagen, previous studies showed that other connective tissue constituents are also argyrophilic, e.g., type I, IV and V collagen, fibronectin and proteoglycans [87,179,191]. Furthermore, in regard to deposition and accumulation of connective tissue in the bone marrow, earlier studies of bone marrow from myelofibrosis patients principally described three stages of myelofibrosis [178,192,193,194,195,196,197,198], suggesting that bone marrow fibrosis progresses from an early stage with only slightly increased fiber network (stage 1) to an advanced stage with pronounced hematopoietic hypocellularity and dense myelofibrosis together with osteomyelosclerosis (stage 3). Using the new well-defined WHO criteria, in the earliest stage of PMF—early prefibrotic myelofibrosis—the reticular fiber network in the bone marrow is normal or only focally slightly increased [199,200,201,202]. Importantly, however, the WHO criteria for the discrimination of ET from prefibrotic PMF have been questioned and not been shown to be reproducible in all studies, leading to a higher proportion of non-classifiable MPNs [203]. Indeed, this discordance calls for other means to differentiate ET from early prefibrotic myelofibrosis. In this context, the measurement of circulating ECM fragments may be useful and needs to be investigated.

Although PMF may evolve from a stage of hypercellularity with minimal fibrosis to an advanced stage with myeloid atrophy and myelofibrosis, this evolution has not been convincingly proven via sequential bone marrow biopsies during the course of the disease. A few studies have attempted systematically to follow the evolution of bone marrow fibrosis, and highly different results were obtained [193,196,197,198,204,205,206,207,208,209]. Hasselbalch and Lisse found a constant bone marrow pattern or regression of bone marrow fibrosis to be the most common [209]. This was also recorded in other series [193,196,204,205,206], even though progressive myelofibrosis was recorded in some studies [198,207,208]. Lennert et al. quantified the number of argyrophilic fibers in bone marrows from patients with various MPNs, including PMF [178]. Although detailed data were not presented, they stated that MPNs progressed from a low to a high fiber content. Furthermore, they observed the fiber content to be higher in patients who had both fibrosis and osteomyelosclerosis [178]. Using digital image processing, no correlation was found in these earlier studies between bone marrow fiber content and the duration of the disease from the time of onset of symptoms [210].

These discordant results have been attributed to several factors, including different criteria for diagnosis. Thus, some studies enrolled patients without classical features of PMF at first biopsy, i.e., increased the amount of reticular fibers in the bone marrow, leucoerythroblastic anemia and splenomegaly [207,208,211]. Today, such patients are classified as early prefibrotic myelofibrosis [199,200,201,202,211], which are characterized by no or minimal “reticulin fibrosis”, no or minor anemia and in most patients, a normal or only slightly enlarged spleen. The distinction between ET and early prefibrotic myelofibrosis has prognostic implications since patients with prefibrotic MF have a poorer prognosis as compared to ET [212,213] but a much better prognosis than classical PMF patients with anemia, large spenomegaly and dense bone marrow fibrosis [214]. Despite the fact that the disease entity of early prefibrotic myelofibrosis has been included in the international WHO criteria since 2016 [215,216,217], and today is used worldwide, concerns have been raised in regard to its relevance as a separate MPN subgroup [203,218,219,220], taking into account the rather low reproducibility of the assessment of bone marrow features, which define early prefibrotic MF, even amongst trained haematopathologists [201,218,219]. For obvious reasons, the patchy nature of the bone marrow lesions in the early MPN stages, even within the same bone marrow biopsy specimen, may be challenging in terms of a precise “general“ description of the bone marrow in MPNs. Therefore, novel tools to assess the stroma changes in MPNs are needed, such as serial measurements of circulating, stage-specific ECM protein fragments, which will be addressed below.

Although progressive myelofibrosis was recorded in most patients with so-called “early myelofibrosis “ in the previous studies [207,208,211], no progression of myelofibrosis was demonstrated in patients who at diagnosis fulfilled the criteria for classical primary myelofibrosis [207,208]. As alluded to above, Wolf and Neiman [205] also emphasized that the absence of detectable progressive myelofibrosis might be related to the patchy nature of the stromal proliferation with a striking variability in the amount of reticulin and haematopoietic cellularity within a given biopsy [221] and most likely also in various parts of the skeleton. Furthermore, reversal of myelofibrosis has been shown to occur in some patients, either spontaneously [222,223], following splenic irradiation [224,225], splenectomy [226,227,228], bone marrow transplantation [229,230,231,232,233,234,235,236], chemotherapy [204,206,237], monotherapy with pegylated interferon-alpha [238,239,240,241], combination therapy with ruxolitinib and pegylated interferon-alpha2 [242,243,244], or immunosuppressive therapy [245,246,247,248,249,250,251], including monotherapy with ruxolitinib [252]. Finally, resolution of bone marrow fibrosis has been observed in patients who enter leukemic transformation [198,204,205].

## 5. Biochemical and Immunohistochemical Studies of Bone Marrow

The total amount of collagen (hydroxyproline) in bone marrow is increased in PMF [253]. In the early phase of the disease, soluble collagen with reducible cross-links appears to prevail, whereas insoluble, stably cross-linked collagen is abundant in advanced disease [253]. In addition, the content of glycosaminoglycans is decreased, particularly in patients with advanced disease [253]. In most patients, urinary hydroxyproline excretion is normal [89]. Immunohistochemical studies have shown normal bone marrow stroma to be composed of interstitial collagen types I and III (III > I) together with basement membrane collagen (type IV) and type V collagen [87,95,254,255]. Collagen types I and III are scattered throughout the bone marrow, whereas type IV collagen is localized discontinuously beneath the lining of sinusoids. [87,95,255]. In addition to type IV collagen, normal bone marrow basement membrane matrix is composed of non-collagenous glycoproteins, including laminin and fibronectin [87,95,180]. Laminin and type IV collagen have been shown to be closely co-distributed in normal bone marrow [87,95].

As mentioned above, in PMF, there is an increased deposition of both interstitial matrix collagens and basement membrane collagens in the bone marrow, together with increased numbers of sinuses and capillaries [87,95,170,173,174,175,176,177,178,255]. The interstitial reticulin network in PMF has been shown to be composed mainly of type III collagen and, in many patients, also mature type I collagen fibers. In addition, fibronectin contributes to the reticulin matrix [179]. The type III collagen fibers have been demonstrated by the application of antibodies against type III procollagen (pN collagen) [86,95], indicating that the propeptide is partly retained on the collagen fibers in the bone marrow.) Particularly increased collagen content, including both newly deposited type III procollagen (pN collagen) and mature collagen fibers, was found in patients with osteomyelosclerosis, which was also associated with the most pronounced neovascularization [95]. The demonstration of partially processed type III procollagen containing its aminoterminal propeptide (= pN collagen) in normal and fibrotic bone marrow is in accordance with studies of other tissues, e.g., skin, in which co-distribution has been demonstrated between type III collagen and type III pN collagen fibrils [256].

Besides fibroblast and endothelial proliferation, another important change in bone marrow in PMF includes the formation of continuous sheets of basement membranes beneath endothelial cells [86,95,255]. This process appears to be very similar to the “capillarization” of liver sinusoids during the development of liver cirrhosis [257]. Despite being discovered for the first time approximately 30 years ago, the pathophysiological implications of altered basement membrane structure in myelofibrosis are, at present, largely unknown. Conceivably, the changes might eventually interfere with the migration of hematopoietic cells from the bone marrow compartment across the marrow-blood barrier into the circulation, allowing stem cells and other immature progenitor cells to enter the circulation prematurely [85].

## 6. Studies of Extracellular Matrix Metabolism

As previously described, the connective tissue responds uniformly to injuries of any kind by distinctive sequential changes in the ECM expression, including oedema formation, angiogenesis and finally, fibrosis, with the deposition of type III collagen in the early phase, mainly as fine fibers, and type I collagen as coarse fibers in the later phase of the lesion (Figure 2). This injury–repair process is qualitatively similar in all organs and is accompanied by the release of various matrix components into the circulation during the synthesis and breakdown of connective tissue constituents at the site of injury. About 30 years ago, radioimmunoassays for the measurement of these connective tissue components in circulation were developed [258,259]. Since then, these methodologies have been extensively elaborated and are today being used to monitor collagen metabolism in various inflammatory and fibrosing disorders, including cancers at the time of diagnosis, during treatment to monitor disease activity and response to treatment and in prognostification as well. In the following, we summarize MPN studies on circulating ECM-associated proteins such as collagen metabolites, matrix metalloproteinases (MMPs) and tissue inhibitors of metalloproteinases (TIMPs) (Table 1).

### 6.1. The Aminoterminal Propeptide of Type III Procollagen (PIIINP)

All fibrillar collagens (e.g., types I and III) are synthesized as procollagens, which contain additional propeptide extensions at both ends. Prior to collagen fiber formation, the procollagens are converted to collagens. During this process, the propeptide regions of the procollagen molecules are cleaved off in the extracellular space and released into the circulation [81,82,83,84,85,86,158,159,260,261,262,263,264,265,266,267,268,269,270,271]. Since the propeptides are released in equimolar amounts to the collagen molecules formed, the measurement of circulating levels of these propeptides is thought to reflect collagen synthesis (Figure 1). Willumsen et al. summarized this process recently for neo-epitop specific type III collagen propeptides (PRO-C3) measured in serum in a fraction of patients with various solid tumors, highlighting the prognostic value in the context of overall survival (see below Future Research Directions) [154].

In previous studies of PIIINP in patients with PMF and related neoplasms [81,83,84,85,86,261,262], the highest serum levels were found in patients with PMF [81,83,84,86,91], in those PV patients transforming into a myelofibrotic stage of the disease [81,83,86], and in patients with chronic myelogenous leukemia (CML) associated with bone marrow fibrosis [84,86]. Although these studies suggested serum PIIINP as a useful noninvasive means to monitor the accumulation of interstitial type III collagen in the bone marrow, several factors should be considered when interpreting serum PIIINP levels. First, the radioimmunoassay for the aminoterminal propeptide of type III procollagen (RIA-gnost, Hoechst, FRG), used in the seminal studies on circulating PIIINP in MPNs, did not exclusively detect the authentic propeptide released during the conversion of procollagen to collagen but measured at least three molecular weight variants of the antigens [263,264]. The main antigen in normal serum is smaller than the authentic aminopropeptide liberated during the conversion of type III pN collagen to type III collagen. These lower molecular weight peptides were shown to probably represent degradation products of the propeptide [265,266], including those related to the degradation of newly synthesized type III procollagen or degradation of that portion of the propeptide retained in situ in the bone marrow. However, in rheumatic diseases, the local degradation of pN-collagen in inflamed tissue has not been considered to contribute significantly to elevated serum aminopropeptide levels [265].

Owing to the heterogeneity of the antigens related to the aminoterminal propeptide of type III procollagen, a next-generation equilibrium RIA for the N-terminal propeptide of human type III procollagen (Farmos Diagnostica, Oulu, Finland) was additionally applied in a series of patients with PMF and related neoplasms [91]. This assay system did not detect the smaller antigen variants, but only antigen forms equal to or larger than the authentic propeptide. A highly significant correlation was found between the two assay systems, which both discriminated between patients with stable and active disease, the highest propeptide levels being recorded in patients with a syndrome of acute myelofibrosis and those with accelerated disease [91]. Furthermore, serum propeptide levels correlated significantly with conventional markers of disease activity (leukocyte count and plasma lactic dehydrogenase), which was also observed in longitudinal studies, where a close co-variation between serum PIIINP and these parameters was recorded.

Analysis of the antigen profile of PIIINP by means of gel filtration in patients with different disease activity showed smaller antigen variants to prevail in patients with stable disease, whereas the intact propeptide dominated in patients with acute or transforming disease. This observation indicated that elevated circulating PIIINP levels primarily reflect de novo synthesis of type III collagen and not increased degradation of the propeptide [91]. Vellenga et al. also found that the antigenicity related to PIIINP was heterogeneous, with at least two main peaks having molecular weights equal to and smaller than the authentic propeptide [261]. Cytotoxic treatment was accompanied by decreasing serum propeptide levels [91]. Taken together, these early observations indicated type III collagen metabolism to be closely linked to clonal myeloproliferation [86,91]. Recently, Willumsen et al. introduced an assay that specifically measures propeptide containing the neo-epitope cleavage site generating the true propeptide (PRO-C3) and, therefore, measures the true formation of type III collagen [105,154] (see Future Research Directions).

Second, while an assay for serum PIIINP mainly measures de novo synthesis of type III procollagen at the time of investigation, the degree of bone marrow fibrosis as assessed by a bone marrow biopsy provides evidence about the preceding net collagen deposition [85,91].

This accumulation may be due to both an increased collagen synthesis and/or impaired degradation of bone marrow collagen. Thus, the observed relationship between increasing degrees of bone marrow fibrosis and increasing serum propeptide levels rather reflects an enhanced myeloproliferative activity and hence ongoing stimulation of bone marrow fibroblasts.

Third, circulating type III procollagen-related antigens can be derived from sources other than the bone marrow, including both the spleen and liver, due to myeloid metaplasia and fibroplasia in these organs [206]. Thus, raised serum PIIINP may theoretically also reflect impaired propeptide metabolism in patients with massive myeloid metaplasia in the liver since serum PIIINP is increased in certain liver diseases [263,271] and the liver is a main site of PIIINP uptake and degradation [269]. Accordingly, extramedullary clonal myeloproliferation may contribute to the increased serum concentration of PIIINP in patients with massive myeloid metaplasia of the liver.

Concerning the relationship between the serum levels of PIIINP and disease activity or disease progression and the degree of bone marrow fibrosis in PMF and related neoplasms [81,86,91,261,262], it is important to realize that serum PIIINP is a marker of current organ fibrogenesis (disease activity), whereas fibrosis in the light microscope reflects the amount of previously deposited collagen (disease stage). This concept is substantiated by studies of liver patients, showing equally raised serum values in patients with fatty liver and inactive cirrhosis [271] and no relationship between serum PIIINP and the severity of liver fibrosis [271]. However, a relationship was found between serum PIIINP and ultrastructural, but not light microscopic fibrosis, which might indicate that raised serum PIIINP levels reflect early collagen formation in the liver [271].

### 6.2. Biomarkers of Type I Collagen Metabolism

Circulating carboxyterminal telopeptide of type I collagen (ICTP) in MPNs as a biomarker of type I collagen degradation has been found to be significantly elevated in patients with MPNs as a group compared with healthy controls [272]. However, subgroup analysis displayed only significantly elevated ICTP levels in patients with PV and MF. Of note, in this study, a significant correlation between the biomarker of collagen synthesis—PIIINP—and ICTP was recorded, reflecting that the aberrant collagen metabolism in MPNs is characterized by both enhanced collagen synthesis and degradation, the integrated signature of “a wound that never heals“ due to persistent release in the bone marrow of several growth factors (e.g., TGF-beta, FGF, and VEGF) from the malignant hyperproliferating myeloid cells in the bone marrow. Importantly, we found normal levels of biomarkers of type I collagen synthesis–carboxyterminal peptide of type I collagen (S-PICP) and aminoterminal propeptide of type I collagen (S-PINP), respectively, indicating that in MPNs, collagen I metabolism is disrupted in favour of enhanced collagen I catabolism. This also highlights the importance of not just measuring any type I collagen fragment, but preferentially, a specific fragment, or neo-epitope, that may provide further and reliable information on disease activity/anabolism versus collagen degradation. In contrast, collagen III formation is increased consequent to sustained repair processes in the bone marrow microenvironment as reflected in the circulation by elevated PIIINP levels [273] and in the bone marrow as increased reticulin fibrosis, being most prominent in patients with PV and MF [95]. The catabolism of type III collagen, as can be measured using fx MMP-degraded type III collagen (C3M), remains to be explored thoroughly in MPNs, but has shown promising results in various solid tumors (see Future Research Directions) [135,274].

As alluded to below, the trigger event behind this phenotype shift in collagen homeostasis with significant positive correlations between plasma soluble urokinase plasminogen receptor (suPAR), and PIIINP, hyaluronan and ICTP levels might be an increased production of matrix-degrading enzymes in the bone marrow [275]. Notably, biomarkers of remodelling (PIIINP, ICTP, hyaluronan) in the bone marrow (and elsewhere) correlated significantly with circulating suPAR, reflecting the activity of the urokinase-type plasminogen activator (uPA) system, which plays an important role in matrix degradation and remodelling [275]. In contrast, another family of extracellular proteolytic enzymes, the matrix metalloproteinases (MMPs), herein 2 and 9) and tissue inhibitors of metalloproteinases (TIMPs), which are also important catalyzers of matrix degradation and remodeling (please, see below) were not correlated with circulating biomarkers of collagen metabolism [275].

### 6.3. Basement Membrane Proteins

The fibrotic process in PMF involves the proliferation of both fibroblasts and endothelial cells [87,95,179,255]. Endothelial proliferation is accompanied by a marked increase in basement membrane matrix components (type IV collagen and laminin), which are deposited as continuous sheets beneath the endothelial cells in the bone marrow [87,95]. The increase in basement membrane structures in myelofibrosis is also reflected in the circulation, where the concentration of both type IV collagen and laminin is increased in a proportion of patients with PMF and related diseases [85,276].

Increased serum levels of type IV-collagen-related antigens have also been found in patients with alcoholic liver disease [271,277], where they reflect type IV collagen metabolism, including the formation of basement membranes beneath the endothelial cells (capillarization of the sinusoids) during the development of liver cirrhosis [277]. This process appears to occur concomitantly with the deposition of collagen in the space of Disse (collagenization) [278], which is also evidenced by a positive correlation between the serum concentration of 7S collagen antigen and PIIINP [271].

Hasselbalch et al. found a similar relationship between the serum concentration of these connective tissue metabolites in PMF, which may actually reflect that the development of bone marrow fibrosis follows a pattern similar to the development of liver fibrosis, being associated with a parallel deposition of interstitial collagen type III and basement membrane material (type IV collagen) [85].

Laminin is a non-collagenous glycoprotein, which, like fibronectin, easily adheres to cell surfaces. Plasma fibronectin has been found to be inversely related to spleen size, the lowest levels being recorded in patients with huge spleens [88]. A similar relationship has been shown between serum laminin and spleen size [276], which may explain that serum laminin values were within the normal range in patients with large spleens despite evidence of active disease. However, normal serum laminin levels have also been recorded in patients with active rheumatoid arthritis, but spleen size was not evaluated in this patient group [279]. In conclusion, interstitial collagen and basement membrane metabolism in PMF are closely interrelated, which may reflect a shared stimulus for fibroblast and endothelial proliferation.

### 6.4. Hyaluronan

Hyaluronan (hyaluronic acid) is a highly hydrophilic glycosaminoglycan and a main component of the ECM amorphous ground substance, being particularly prevalent in soft connective tissues [171,280]. Elevated serum hyaluronan (HYA) levels have been found in various inflammatory connective tissue diseases, including rheumatoid arthritis [281] and scleroderma [282]. Certain liver diseases are also associated with elevated serum HYA [283]. Hyaluronan has been shown to be synthesized by fibroblasts upon various stimuli, including platelet-derived growth factor [282], which is also assumed to be involved in the pathogenesis of bone marrow fibrosis in PMF [284,285,286,287].

Raised serum HYA concentrations have been reported in a proportion of patients with PMF [96]. Serum HYA concentrations in patients with acute disease were significantly higher than those recorded in patients with chronic disease. Furthermore, serum HYA correlated significantly with serum PIIINP and with the leucocyte count, a close co-variation being found between serum HYA, serum PIIINP and the leucocyte count with parallel changes in patients, transforming into an accelerating disease phase and in those receiving intensive chemotherapy [96].

Several explanations may be offered for the relatively modest changes in serum HYA in patients with MPNs. The metabolism of HYA may be altered in patients with splenomegaly, implying a rapid turnover of HYA. Normally, HYA is rapidly cleared from the circulation by the liver endothelial cells [288,289,290,291,292], but some circulating HYA is also taken up by the spleen and other lymphoid organs [292].

Thus, the possibility exists that splenic uptake of HYA is enhanced in patients with splenomegaly, explaining the virtually normal serum HYA concentrations in patients with large spleens, despite evidence of connective tissue activation as reflected in elevated serum PIIINP concentrations [96].

The origin of excess HYA in some myelofibrosis patients remains to be clarified. Since both fibroblasts and endothelial cells have the capacity to synthesize HYA [293], raised HYA levels in the circulation may reflect the stromal cell reaction to clonal myeloproliferation. Alternatively, excess HYA may be a marker of the malignant clone itself, having the potential for HYA synthesis and production. Thus, a heterogeneous group of glycosaminoglycans has been identified in human neutrophil granules and in Auer rods of leukemic myeloblasts [294]. In addition, proteoglycan synthesis has been demonstrated in hematopoetic progenitor cells [295]. Platelet-derived growth factor (PDGF) is implicated in the pathogenesis of PMF [284,285,286,287]. Since PDGF and other growth factors are able to stimulate HYA production [282], it is intriguing to consider the possibility that excess HYA in the circulation in PMF is caused by the intramedullary release of growth factors from the megakaryocyte cell lineage with the ensuing stimulation of HYA production from bone marrow fibroblasts and endothelial cells. Finally, impaired metabolism of HYA owing to myeloid metaplasia in the liver might contribute to elevated HYA levels in this patient group [283]. The temporal relationship between the above connective tissue components, which are similar to those observed in inflammatory connective diseases, lends support to the concept that repair processes are taking place in the bone marrow in response to clonal myeloproliferation. Thus, the co-variation of serum HYA and serum PIIINP conceivably reflects a common underlying mechanism of connective tissue activation in PMF and related diseases. However, considering the relatively small changes in serum HYA, the clinical utility of single determinations of serum HYA is limited.

### 6.5. Fibronectin

As mentioned above, fibronectin is an important connective tissue constituent, which is excessively deposited in the bone marrow in PMF [179,180], mainly with a perivascular distribution which supports that the elevated levels in peripheral blood originate from the bone marrow compartment. Low levels of circulating fibronectin levels in patients with myelofibrosis have been associated with large spleens [88] or binding to circulating immune complexes [90,92]. Alternative splicing of the FN gene results in the generation of protein variants that contain the additional isoforms—extra domain A of FN (EDA) and extra domain B of FN (EDB); FN (EDA) and FN (EDB) are associated with tissue remodeling, fibroblast differentiation, inflammation, and tumor progression [296]. Highly intriguingly, EDA-FN has been shown to sustain megakaryocyte proliferation and induces a proinflammatory phenotype in bone marrow cell niches [297]. Based upon these observations circulating plasma levels of EDA-FN have been measured in patients with primary myelofibrosis in whom the highest levels of plasma EDA-FN were recorded in patients with a homozygous *JAK2V617F* genotype. Furthermore, increased EDA-FN levels are associated with anemia, increased high-sensitivity C-reactive protein and bone marrow fibrosis. Since elevated plasma EDA-FN at diagnosis was also found to be a predictor of large splenomegaly (over 10 cm from the left costal margin), it was concluded that plasma EDA-FN might be a new marker of disease progression and a novel target for the treatment of splenomegaly [298].

### 6.6. Metalloproteinases (MMPs) and Tissue Inhibitors of Metalloproteinases (TIMPs)

The abnormal accumulation of ECM in any organ is determined by the balance between matrix anabolic processes and the activity of matrix-degrading enzymes (e.g., MMPs, cathepsins) and their inhibitors (e.g., TIMPs and alpha-2 macroglobulin). These processes can be studied by measuring the plasma concentrations of MMP family members (e.g., MMP-1, MMP-2, MMP-3, MMP-9) and tissue inhibitors of MMPs (TIMPs) (e.g., TIMP-1). If the normally delicate balance in tissue modelling or remodelling after tissue injury of any kind is disturbed in favour of a profibrotic state, this imbalance may be reflected by increased circulating levels of anabolic matrix metabolites and/or decreased MMPs and/or increased levels of TIMPs. Studies on plasma levels of MMPs and TIMPs in MPNs [275,299,300] have shown decreased levels of several MMPs and increased TIMPs, in particular in patients with MF, in whom the TIMP-1/MMP-9 ratio was found to be significantly higher as well [275]. Furthermore, whole-blood gene expression profiling studies have shown several MMPs and TIMP3 to be deregulated in MPNs with the significant upregulation of MMP1, MMP7, MMP8, MMP9, MMP11, MMP12, MMP14 and TIMP3 [301].
cancers-15-04323-t001_Table 1Table 1Studies on circulating extracellular matrix-related proteins in myelofibrosis and related neoplasms.BiomarkerNo. PatientsSummary of ResultsConclusions/CommentsRefs.S-PIIINP441S-PIIINP values were increased in PV with even more elevated levels in post-PV-MF and strikingly elevated in patients with severe myelofibrosis [81].S-PIIINP associates significantly with the extent of reticulin fibrosis.S-PIIINP is a quantitative marker for myelofibrosis [81];[81,83,84,85,86,96,98,261,262,268,272,276]S-PIIINP values were normal in patients without reticulin fibrosis; increased in PV and MF.S-PIIINP values above 25 ng/mL associated with MF of recent onset (less than or equal to 2 years) and values below 25 ng/mL with MF of more than 4 years’ duration [83].S-PIIINP is a non-invasive method for accurate assessment of bone marrow fibrosis.S-PIIIINP may be used to evaluate the efficacy of antifibrosing agents [83].S-PIIINP values were increased in PV and related to degree of reticulin fibrosis.S-PIIINP values were increased in spent phase of PV only treated with phlebotomy.S-PIIINP values were increased in patients transforming into post-PV-MF and increased in PMF [86].S-PIIINP is higher in myelofibrosis of recent onset (less than 2 years) than in myelofibrosis longer than 2 years.S-PIIINP is stable in PV patients treated with 32P or hydroxyurea [86].S-PIIINP values were increased in PV and even more in patients with TMD and MF; S-PIIIINP values were virtually normal in OMS (deposition of type I collagen) [84].S-PIIINP is a useful indicator of disease activity in MPNsS-PIIINP positively correlates to the degree of reticulin fibrosis.Near normal S-PIIINP values in OMS likely reflect stable disease without concurrent type III collagen synthesis. [84].S-PIIINP values were normal or elevated in PV and TMD.S-PIIINP values were normal or even low levels in OMS.S-PIIINP values were increased in MF and in CML associated with bone marrow MF. S-PIIINP and S-Type IV collagen correlated significantly with each other and with the leucocyte count [85].S-PIIINP is a useful indicator of disease activity in MPNs.Normal and even low S-PIIINP values in OMS may reflect stable disease without ongoing type III collagen synthesisInterstitial type III collagen and basement membrane metabolism are closely related [85].S-PIIINP values were strongly raised in MPNs.All three biomarkers (S-PIIINP, S-PICP and S-Laminin ) were significantly elevated in patients with active/transforming disease [98].S-PIIINP is a useful indicator of disease activity in MPNs.S-Laminin and S-PICP do not offer offer any advantage over S-PIIIP.Interstitial type III collagen and basement membrane metabolism are closely related [98].S-PIIINP values were slightly elevated in patients with stable disease and highly elevated in patients transforming into myelofibrosis.S-PIIINP values covariated closely with S-laminin, the leucocyte count and LDH [276].S-PIIINP is a useful indicator of disease activity in MPNs.Interstitial type III collagen and basement membrane metabolism are closely related [276].S-PIIINP values correlated significantly with the leukocyte count and with S-HU.S-PIIINP values decreased during cytotoxic treatment in concert with declining leukocyte counts and S-HU [96].S-PIIINP is a useful indicator of disease activity in MPNs.S-PIIINP may be useful in monitoring the efficacy of cytotoxic treatment in terms of inhibiting development and progression of bone marrow fibrosis [96].S-PIIINP values were increased in PMF.S-PIIINP values were only increased slightly in patients with stable disease (n = 3) compared to the single patient with more active disease.S-PIIINP values declined during treatment with acetylsalicylic acid (ASA), although normalization did not occur.Using gel filtration analysis the antigens related to S-PIIIINP were found to be heterogenous with at least two peaks, exhibiting molecular masses equal to and smaller than PIIINP [261].This study includes only four patients and accordingly does not allow robust conclusions [261].Studies on the impact of ASA and anti-inflammatory treatment upon type III collagen metabolism are needed [261].S-PIIIP values were elevated in PMF.S-PIIIP values were normal in younger patients, having higher Hb- and platelet counts and lower S-ferritin values platelet count.S-PIIIP values were significantly higher in patients with active disease (fever, sweating, weight loss) than in patients with non-active disease S-PIIIP values correlated with decreasing Hb-concentration and platelet count and increasing WBC, serum ferritin and number of transfusions (univariate analysis).S-PIIINP values correlated independently with increasing WBC, serum ferritin and age (multivariate analysis).S-PIIINP values did not associate with morphometric grading of bone marrow fibrosis, megakaryocyte number, or lymphoid infiltration [262]S-PIIIP values in PMF correlates more with overall disease activity than with the extent of bone marrow fibrosis [262].The association between normal S-PIIINP and lower S-ferritin values in younger patients with higher HB-concentrations may likely reflect that S-PIIINP also is a biomarker of the chronic inflammatory state in PMF.Studies on the associations between S-PIIINP, biomarkers of chronic inflammation (e.g., CRP, ferritin, inflammatory cytokines), bone marrow megakaryocyte morphology and bone marrow fibrosis and the impact of cytotoxic (HU) or stem-cell targeting therapy ( pegylated interferon-alpha2 ) as monotherapies or in combination (e.g., with JAK1-2 inhibitor) are needed.S-PIIINP values were elevated in PMF,S-PIIINP values decreased during treatment with anthracycline, which was given due to accelerated phase disease [268].S-PIIINP is a valuable biomarker in PMF.Cytotoxic treatment lowers elevated S-PIIINP values [268].S-PIIINP values were elevated in MPNs.S-PIIINP values were highest in patients with MF.S-PIIINP and S-ICTP correlated significantly.S-PIIIINP and P-suPAR correlated significantly.S-PIIINP values did not correlate with P-MMP-2 and MMP-9 [272].S-PIIINP is a useful indicator of disease activity in MPNs.Type III and type I collagen metabolism are closely associated, reflecting concurrent type III synthesis (PIIINP) and type I degradation (ICTP).Elevated S-ICTP values in MPN may not only reflect type I collagen degradation in the bone marrow but also increased bone resorption.Enzymes of the uPA system might participate in the bone marrow remodelling processes characteristic of MPN [272].S-PICP26S-PICP values were slightly elevated in MPNs, reflecting increased type I collagen synthesis.S-PICP values were significantly elevated in patients with active/transforming disease.S-PICP and S-laminin P1 values showed a strong correlation [98].S-PICP values do not offer any advantage over S-PIIIP for monitoring of disease activity.Increased type I collagen synthesis associates with progressive disease.Interstitial type I collagen and basement membrane metabolism are closely related [98].[98]S-ICTP50S-ICTP values were elevated in MPN.S-ICTP values were only significantly higher among MF and PV patients.S-ICTP and S-PIIINP values correlated significantly.S-ICTP and P-suPAR values correlated significantly.S-ICTP and P-MMP-2/MMP-9 values did not correlate significantly.Elevated S-ICTP values in MPNs reflect ongoing type I collagen degradation.Elevated S-ICTP values may not only reflect enhanced type I collagen degradation in the bone marrow but also increased type I collagen degradation in bone tissue (increased bone resorption).Increased bone resorption with the development of osteopenia/osteoporosis may be driven by chronic inflammation in MPNs. In this regard, the significant correlation between S-ICTP and P-suPAR may also reflect the chronic inflammatory state and not only the involvement of these biomarkers in the bone marrow remodeling processes in MPNs.[272]S-TIVC41S-TIVC values were normal or elevatedin PV and TMD.S-TIVC values were elevated in MF andin CML associated with bone marrow MF.S-TIVC and S-PIIINP correlated significantly and with the leucocyte count.Measurement of type IV collagen provides a noninvasive means for following the accumulation of basement membrane collagen in the bone marrow in patients with MPN.S-TIVC associates with disease activity as assessed by the leukocyte count.Interstitial (type III collagen ) and basement membrane metabolism (type IV collagen ) are tightly associated processes in MPNs.[85]S-Laminin158S-Laminin1 values were slightly elevated in MPNsS-Laminin1, S-PIIINP and S-PICP values were significantly elevated in patients with active/transforming disease.S-Laminin P1 and S-PICP levels showed a strong correlation [98].S-Laminin did not appear to offer any advantage over S-PIIIP for the monitoring of disease activity.Basement membrane (laminin) and interstitial collagen ( PIIINP, PICP) metabolism are closely related in MPNs [98].[98,276]S-Laminin1 values were slightly elevated in patients with stable disease.S-Laminin1 values were highly elevated in patients with progressive disease transforming into myelofibrosis.S-Laminin1 covariated closely with S-PIIINP, the leucocyte count and LDH [276].S-Laminin1 values were significantly lower in patients with a huge spleen as compared with patients, having a normal spleen size or previously being splenectomized.The above observation may reflect that the aminin uptake/degradation is increased in the enlarged spleen in MPNs.S-HYA59S-HYA values were normal in patients with stable disease and increased in patients with active disease. S-HYA values correlated significantly with the leukocyte count and with S-PIIINP.S-HYA values decreased during cytotoxic treatment in concert with declining leukocyte counts and S-PIIINP.S-HYA values displayed only slight changes in MPNs with with frequent overlaps between patient categories and HC. The clinical utility of S-HYA may be restrained, although sequential measurements of S-HYA may provide a reflection of the MPN process in individual patients.[96]P-Fibronectin69P-Fibronectin values were normal in ET.P-Fibronectin values were significantly reduced in PV and MF.P-Fibronectin values were lowest in in patients with marked splenomegaly.P-Fibronectin values rose from less than 100 mg/L to 177 mg/L after splenectomy in a patients with MF [82].Low P-fibronectin values in MPNs may be attributed to increased consumption of P-Fibronectin in the expanded mononuclear phagocyte system in the liver and spleen, reduced hepatic synthesis, and/or fibronectin taking part in the clearance of circulating immune complexes.Low P-Fibronectin values in patients with MPNs may contribute to an increased risk of infections.[82,88,90]P-Fibronectin values were significantly lower in patients with PMF.P-Fibronectin values in MF patients differed significantly, when compared with patients with PV, TMD or CML.P-Fibronectin values were lowest in patients with large spleens [88].Low P-fibronectin concentrations in splenomegalic patients may be due to enhanced consumption of the opsonin in the expanded splenic mononuclear-macrophage system [88].P-Fibronectin values correlated inversely with CIC, which were highly elevated in 11 of 20 with MF secondary to MPN.The CIC contained fibronectin, IgG and C3.P-Fibronectin levels increased after therapeutic plasmapheresis, which efficiently removed CIC [90].The findings suggest that fibronectin as a major non-specific opsonin is important for the normal clearance of immune complexes [90].P-Fibronectin (EDA)122P-EDA FN values were significantly elevated in PMF as compared to HCs.P-EDA F values did not differ betweenPV/ET patients and HCs.P-EDA FN values differed among patients with different degrees of BM fibrosis with a trend towards increasing P-EDA FN levels with increasing BM fibrosis grades (not significant).P-EDA FN differed significantly between patients with pre-fibrotic myelofibrosis (BM fibrosis grade 0) + those with early myelofibrosis (BM fibrosis grade 1) as compared to those with BM fibrosis grade 2 + those with BM fibrosis grade 3 (advanced fibrosis) [297].Patients with PMF exhibited higher levels of the EDA FN isoform as compared to HCs.[297,298]P-EDA FN values were higher in patients with a homozygous JAK2V617F genotypeIncreased P-EDA-FN values were associated with anemia, elevated high-sensitivity C-reactive protein, bone marrow fibrosis and splanchnic vein thrombosis at diagnosis.Elevated P-EDA-FN at diagnosis was a predictor of large splenomegaly [298].P-EDA-FN in primary myelofibrosis may represent a marker of disease progression, and a novel target to treat splenomegaly [298].P-YKL-4048P-YKL-40 values were significantly elevated in PMF vs. control subjects.P-YKL-40 values were increased from ET over PV to PMF [302].P-YKL-40 may be a novel biomarker of chronic inflammation, tissue remodelling and atherosclerotic inflammation in MPN[302,303].[302,303]P-YKL-40 values were significantly elevated in PMF vs. control subjects.P-YKL-40 values were increased from ET over PV to PMF [302].P-YKL-40 might be a novel marker of disease burden and progression in MPN[303].S-YKL-40111S-YKL-40 values were significantly higher in post-ET MF, PV, post-PV MF and PMF patients, when compared to HC.S-YKL-40 values were associated with biomarkers of an increased inflammatory state (higher C-reactive protein, poor performance status, presence of constitutional symptoms and cardiovascular risk factors).Higher S-YKL-40 values in MF patients were also associated with blast phase disease, lower hemoglobin and higher Dynamic International Prognostic Scoring System score. Higher S-YKL-40 values were independently associated with an increased risk of thrombosis and impaired survival in MF patients [304].Higher S-YKL-40 might have a pathophysiological role in disease progression and thrombosis development.Assessing S-YKL-40 could help in identification of ET and PV patients at a high risk of future cardiovascular events and has a good potential for improving prognostication of MF patients [304].[304]S-CHIT191S-CHIT1 values were significantly higher in PV and post-PV myelofibrosis transformation (post-PV MF).S-CHIT1 values were not significantly higher in ET, post-ET MF transformation, and PMF patients, when compared to healthy controls.S-CHIT1 values in PV were positively correlated with hemoglobin, hematocrit, absolute basophil count and the presence of reticulin fibrosis in the bone marrow.A positive correlation between S-CHIT1 and the hemoglobin, hematocrit, and absolute basophil count in PV might imply macrophages closely related to clonal erythropoiesis as cells of CHIT1 origin. A positive association between S-CHIT1 and reticulin fibrosis might indicate its potential role in PV progression.S-CHIT1 might a circulating biomarker of macrophage activation with an important role in inflammation-induced tissue remodeling and fibrosis in PV.[305]P-Pentraxin-3(P-PTX3)244/477/140P-PTX3 and P-hs-CRP were measured in 244 consecutive ET and PV patients.After a median follow up of 5.3 years (range 0–24), 68 CV events were diagnosed.Major thrombosis rate was higher in the highest hsCRP and lower at the highest PTX3 levels. These associations remained significant in multivariate analyses [306].P-hs-CRP and P-PTXT3 independently and in opposite ways modulate the intrinsic risk of CV events in patients with MPN [306].[306,307][308]P-PTX3 levels in 477 ET and PV patients were significantly increased in carriers of homozygous *JAK2V617F* mutation compared to all the other genotypes and triple negative ET patients, while hs-CRP levels were independent of the mutational profile.The risk of hematological evolution and death from any cause was about 2- and 1.5-fold increased in individuals with high PTX-3 levels, while the thrombosis rate tended to be lower.High hs-CRP levels were associated with risk of haematological evolution, death and also major thrombosis.After sequential adjustment for potential confounders (age, gender, diagnosis and treatments) and the presence of *JAK2V617F* homozygous status, high hs-CRP levels remained significant for all outcomes, while *JAK2V617F* homozygous status as well as treatments were the factors independently accounting for adverse outcomes among patients with high PTX3 levels [307].The *JAK2V617F* mutation influences MPN-associated inflammation with a strong correlation between allele burden and PTX3 levels. P-hs-CRP and P-PTX3 might be of prognostic value for patients with ET and PV, but their validation in future prospective studies is needed [307].P-PTX3 values were significantly higher in PMF patients than in HC.High PTX3 values (≥70 ng/mL) associated with an unfavourable overall survival.P-PTX3 values independently predicted PMF patients’ overall survival.P-PTX3 values correlated with parameters of tumor burden, including total leucocyte count, mutated JAK2 allele burden, lactate dehydrogenase levels, and spleen size [308].PTX3is released from macrophages and endothelial cells, and promotes the transition of monocytes to fibrocytes.P-PTX3 levels constitute an independent indicator of disease burden, clonal expansion and overall survival in patients with PMF.Monitoring of PTX3 plasma levels might be a useful tool in clinical decision making [308].P-suPAR50P-suPAR correlated significantly with serum markers of collagen metabolism (S-PIIINP and S-ICTP) [272]Enzymes of the uPA system might participate in the bone marrow remodelling processes characteristic of MPN [272].[272,273]P-suPAR values were significantly higher in MPN patients.P-suPAR values differed significantly between MPN-subgroups, the highest levels being found in patients with MF and PV.P-suPAR values were only significantly increased in PV and MF patients.P-suPAR significantly correlated to P-LDH.P-uPA did not differ between patients and controls [273].Increased P-suPAR levels in patients with MPN may reflect increased uPAR production in the bone marrow, leading to enhanced bone marrow remodeling [273].P-TIMP-1P-MMP-1P-MMP-2P-MMP-3P-MMP-967P-MMP-3 levels were decreased in patients with advanced MF.P-MMP-1, P-MMP-2, and P-MMP-9 levels were not significantly different from HC.P-TIMP-1 levels were elevated in ET, PV and MF patients and in particular in advanced MF.P-MMPs levels were not elevated in ET and PV patientsThe abnormal accumulation of ECM is dependent upon the balance between matrix metalloproteinases (MMPs) and tisse inhibitors of metalloproteinases (TIMPs).Accumulation of connective tissue in the bone marrow is associated with reduced MMP activity together with increased TIMP-1 activity, which may be important in fibrosis formation in the bone marrow in MPNs.[299,300]P-TIMP-150Plasma levels of total-, free- and complexed TIMP-1, TIMP-2, MMP-2 and MMP-9 were measured in 50 patients with MPN.P-TIMP-1 levels were significantly higher in MPN patients.P-TIMP-1 levels significantly correlated with P-suPAR and P-uPAR.P-TIMP-2 and P-MMP-2 levels did not differ beween patients and controls.P-TIMP-1 and P-TIMP-2 levels correlated significantly.P-MMP-9 levels significantly higher among PV patients.P-TIMP-1/MMP-9 ratio was significantly higher in patients with MF.The family of MMPs and TIMPs facilitate and inhibit matrix degradation processes, respectively.A disturbed TIMP-1/MMP ratio may reflect an imbalance of the extracellular homeostasis towards an increased matrix deposition promoting fibrosis.[275]U-HYPRL47U-HYPRL was normal in 16 patients with PMF and in 5 out of 6 patients with acute myelofibrosis. In patients with OMS (n = 8) values for U-HYPRL were insignificantly higher than those PMF.U-HYPRL increased in 10 patients (1 AMF patient, 3 OMS patients and 6 patients with CML in the accelerated phase of the disease). All but 1 of these patients had been treated, or were being treated, with cytotoxic agents at the time of investigation [89].The findings of normal U-HYPRL may be explained by impaired degradation of bone marrow collagen which, together with enhanced collagen synthesis from bone marrow fibroblasts, accounts for progressive accumulation of connective tissue in the bone marrow in myelofibrosis patients.This process is influenced by cytotoxic treatment as reflected in increased urinary hydroxyproline excretion in those patients receiving cytotoxic agents [89].[80,89]U-HYPRL was normal in PMF patients.U-HYPRL was increased in patients with metastasis, the highest levels being recorded in those with concomitant bone marrow fibrosis [80].The result suggests differences in the pathogenesis of “MPN-myelofibrosis” (normal U-HYPRL) as compared to myelofibrosis consequent to bone marrow metastasis (increased U-HYPRL) [80].**Abbreviations**: S-PIIINP = Serum N-terminal propeptide of type III procollagen (type III collagen synthesis); S-PICP = Serum-procollagen type I carboxyterminal extension peptide (type I collagen synthesis); S-ICTP = Carboxy-terminal telopeptide of type I collagen (type I collagen degradation; bone resorption marker); STIVC = S-Type IV Collagen; S-HYA = Serum Hyaluronan; U-HYPRL = Urinary hydroxyproline; P-Fibronectin = plasma fibronectin; P-EDA Fn = extra-domain A fibronectin (EDA-FN), P-YKL-40 = Plasma YKL-40; S-CHIT1 = Serum chitotriosidase activity; P-PTX3 = Plasma Pentraxin-3; CIC = hs-CRP = high-sensitivity C-reactive protein; P-suPAR = plasma soluble form of urokinase plasminogen activator (uPAR); P-uPAR = Plasma urokinase plasminogen activator (uPAR); P-TIMP-1 = plasma tissue inhibitors of metalloproteinase -1; P-MMP = plasma metalloproteinase; circulating immune complexes; PV = polycythemia vera; PMF = primary myelofibrosis; MF = myelofibrosis; TMD = transitional myeloproliferative disorder (between PV and myelofibrosis); OMS = osteomyelosclerosis; CML = chronic myelogenous leukemia; HC = healthy controls; WBC = white blood cell count; Refs= reference number.

## 7. Reasons to Revisit Blood-Based Extracellular Matrix Biomarkers in Patients with Myeloproliferative Neoplasms

There are several rationales for reconsidering circulating ECM fragments as biomarkers in patients with MPNs.

First, today MPNs are widely recognized as chronic inflammatory neoplasms, which have been described as “A Human Inflammation Model “and“ A Human Inflammation Model for Cancer Development” [20,21,22,23,24,25,26,27,28,68,73,74,75]. As acquired stem cell neoplasms, driven by inflammatory somatic mutations, they have also been conceived as “wounds in the bone marrow, that never heal” implying that the medullary stroma undergoes persistent repair and regeneration in response to clonal expansion, ultimately leading to bone marrow fibrosis and eventually leukemic transformation.

Second, MPNs develop in a biological continuum spanning decades from the earliest cancer stages—ET and PV—to the advanced myelofibrotic stage, which is characterized by dense collagenization of the bone marrow consequent to a persistent release of growth factors (e.g., PDGF, TGFbeta, VEGF) from rapidly proliferating and dysplastic megakaryocytes [2,4,22,71,72,73,74,75,76,77,78,79]. As such, MPNs constitute a unique model to study circulating ECM biomarkers in different stages of chronic MPN blood cancers, in which chronic inflammation is a major driving force.

Third, MPNs are associated with increased cardiovascular disease burden, including hypertension, risk of stroke and myocardial infarction, ischemic heart failure, and aneurysms [11,12,13,14,15,16,17,18,19,20,21,23]. Patients with MPNs likely develop atherosclerosis earlier due to a chronic systemic inflammatory response similar to premature atherosclerosis development in other inflammatory diseases, such as type II diabetes mellitus, rheumatoid arthritis, psoriasis, and inflammatory bowel diseases, Crohns’ disease and ulcerative colitis [20,21,23]. It is intriguing to speculate if heart failure, in particular heart failure with preserved ejection fraction (HFpEF) is under-recognized in MPNs, considering that HFpEF has been reported in patients with other chronic inflammatory diseases, such as hypertension and type II diabetes mellitus. This is even more pertinent, considering that circulating ECM biomarkers have shown promise as predictive markers of cardiac fibrosis in HFpEF (please see below Future Research Directions) [309,310,311,312,313,314,315,316,317].

Fourth, MPNs are associated with an increased risk of age-related macular degeneration (AMD), which develops much earlier than in the background population [37,38,39,40,41,42]. AMD is characterized by pronounced neovascularization, which may impact levels of circulating ECM biomarkers of increased basement membrane turnover, in view that the retina has the highest blood perfusion rate of all organs.

Fifth, MPNs are associated with an increased risk of fractures, which likely is due to inflammation-mediated osteopenia [43,44,45,46,47,48,49,50]. Circulating ECM markers, reflecting bone resorption (e.g., CTX-I (C-terminal telopeptide of type-1 collagen), and MMP-3 (matrix metalloproteinase-3)) may emerge as complementary tools to the conventional dual X-ray absorptiometry (DEXA) scans of the skeleton for detection of osteopenia in patients with MPNs.

Sixth, MPNs are associated with an increased risk of chronic kidney failure [55,56,57,58,59,60,61,62,63,64,65], which is considered to develop as a consequence of a chronic inflammatory state with ensuing renal impairment. It is intriguing to consider if “MPN-glomerulopathy“ actually develops due to fibrogenesis in the kidneys as a consequence of the intramedullary release of growth factors—mitogenic for fibroblasts and endothelial proliferation (e.g., PDGF, TGGbeta, and VEGF). In this context, elevated circulating ECM biomarkers may come up as an integrated signature of chronic multiorgan inflammation and multiorgan fibrosis, including fibrosis in the kidneys [103].

Seventh, MPNs are associated with an increased risk of second cancers, including lung cancer and urinary tract cancers [29,30,31,32,33,34]. Circulating ECM biomarkers have been shown to be elevated in a large number of solid tumors [102,104,105,106,107,108,109,110,111,112,113,114,115,116,117,119,120,121,122,123,124,125,126,127,129,130,131,132,133,134,135,136,137,138,139,140,141,142,143,144,145,146,147,148,149,150,151,152,153,154,155,156,157]. Accordingly, by serial measurements of ECM markers development of a second cancer may be detected earlier than presently in patients in whom the MPN appears to be well-controlled and in whom ECM marker levels accordingly are expected to be within the normal range.

Eighth, immunomodulating treatment with pegylated interferon-alpha2 (IFN) is highly efficacious in patients in the early MPN stages—ET and PV—and early hyperproliferative myelofibrosis [238,239,240,241,244] but it is of no or minor benefit in patients with advanced myelofibrosis, in whom TGF-beta and VEGF are abundantly expressed in the bone marrow [70,72,73]. However, in a preclinical study, a TGF-β1/β3 protein trap molecule (AVID200) has been shown to have favourable impacts on both hematopoiesis and bone marrow fibrosis in myelofibrosis. Studies in patients with solid tumors have shown that excessive TGF-beta signaling and a collagen-rich peritumoral stroma (desmoplasia) may interfere unfavourably with the interaction between T cells and tumor cells, thereby contributing to resistance mechanisms via immune exclusion [318,319]. Levels of circulating ECM biomarkers have been shown to be correlated with clinical outcomes after PD-1/CTLA-4 inhibition in patients with metastatic melanoma [125,141]. Similarly, ECM biomarkers may be not only useful in predicting response to IFN-alpha2 in MPNs but also in predicting response to PD-1 inhibition in myelofibrosis patients and in response to vaccination as well. Thus, most recent vaccination studies in MPNs have failed in terms of impacting biomarkers of disease activity (leukocyte and platelet counts, *CALR* and *JAK2V617F* allelic burdens), although strong immune responses were recorded in a proportion of the patients [320,321]. It is intriguing to consider if the T-cells are trapped in the fibrosing stroma in patients with myelofibrosis or if the malignant cells are partially protected by the surrounding fibrous stroma against attacks by the immune system (e.g., cytotoxic T-cells, and NK cells). If so, measurements of elevated circulating ECM fragments, both in terms of individual ECM fragments, but perhaps also a particular ECM protein signature, might provide new insights into the mechanisms behind the efficacy or lack of efficacy of immune therapy in MPNs, including immune escape mechanisms.

Ninth, in theory, circulating ECM biomarkers may turn out to become valuable future diagnostic tools to differentiate genuine ET from prefibrotic myelofibrosis, to monitor the transition of prefibrotic MF into classic MF (post-ET MF) [212,213,214,215,216,220], PV transition towards myelofibrosis (post-PV myelofibrosis), and ultimately, leukemic transformation. All these stages are characterized by distinct patterns of several inflammatory biomarkers, including C-Reactive Protein (CRP) [306,307,322], which are conceivably associated with the profiles of single or clusters of circulating ECM components.

Tenth, in the new era of MPNs with minimal residual disease (“operational cure “) as a novel treatment goal [23,36,240,241,242,243,244], implying normal blood cell counts, low-burden *JAK2V617F* (below 1–5%) and a normal bone marrow without fibrosis after long-term treatment with stem-cell-targeting therapy with IFN monotherapy [23,36,241,244] or in combination with JAK1-2 therapy (COMBI) [240,242,243,244], circulating ECM biomarkers may be useful in monitoring early bone marrow stroma changes to increased clonal expansion otherwise remaining undetected by peripheral blood cell counts and accordingly missing an opportunity for the early reinstitution of IFN to halt clonal expansion.

Eleventh, ultimately, circulating ECM biomarkers may assist in predicting overall survival in MPNs by providing an integrated signature of the final outcome of chronic systemic inflammation, e.g., multiorgan failure due to fibrosis in the heart, lungs, kidneys and bone marrow.

Twelfth, most previous studies on circulating ECM fragments in the context of MPNs have focused on major collagens such as collagen type I, III and IV. In recent years, circulating neo-epitope biomarkers have emerged to also quantify the minor and less-well-understood collagens in serum and for which an association with tumor fibrosis and poor prognosis has been indicated, for example, fragments from type XI and XXII collagen measured in serum from patients with pancreatic cancer (a very stroma rich solid tumor type) [146,157]. Such tools may aid in our understanding of bone marrow fibrogenesis as well.

## 8. Perspectives and Future Research Directions

Although previous studies have shown that the serum concentrations of PIIINP, laminin and HYA to co-variate with conventional markers of disease activity, reflecting the close association between fibrogenesis in the bone marrow and hyperactive myeloproliferation [81,83,84,85,86,87,91,97,98], only serum PIIINP seemed to be useful in clinical settings. Since the early studies on serum PIIINP in MPNs, measurement of this type III collagen protein using neo-epitope technology such as PRO-C3 has emerged as a reliable indicator of fibrogenetic activity and prognosis across a range of solid tumors [154]. In addition, other sensitive and reproducible assays have been developed, opening avenues for studies of several novel biomarkers of ECM origin in MPNs. Considering that chronic inflammation has been identified as an important triggering mediator for clonal expansion from the CHIP stage towards overt MPN development but also has a great impact on disease progression in the MPN disease spectrum [18,19,20,21,22,23,24,25,26,27,73,74,75], it seems timely to rethink possibilities and perspectives in the area of circulating ECM marker research [100,101,116,150].

By obtaining further insights into the ECM metabolism in MPNs, there are several perspectives present in such studies, including the introduction of circulating ECM-derived peptides as serological markers singly or as patterns to distinguish between, e.g., genuine ET and prefibrotic myelofibrosis [199,200,212,213,214,216,218,219,220] and to follow the impact of different potentially disease-modifying treatment modalities, such as IFN [238,239,240,241,244] and JAK 1-2 inhibitors [78,252], both as monotherapies and in combination [240,241,242,243,244], since both agents have been shown to induce regression of bone marrow fibrosis in MPNs [238,239,240,241,244]. Additionally, future research should include the measurement of circulating ECM biomarkers in clinical trials by using novel agents that directly target the fibrotic process in the bone marrow [78].

### 8.1. Zinpentraxin alfa

Zinpentraxin alfa is a recombinant human form of pentraxin 2(PTX-2) and seems to be particularly promising. PTX-2 is a regulator of the inflammatory response to tissue injury and fibrosis [78,323,324,325]. PTX-2 binds to damaged cells and facilitates their safe removal in a nonimmunogenic manner [78,323]. By binding to Fcγ receptors and dendritic cell-specific intercellular adhesion molecule-3-grabbing non-integrin (DC-SIGN), PTX-2 facilitates immunoregulatory and phagocytic polarization of monocytes, thereby inhibiting their differentiation into fibroblast-like, collagen-producing cells, named fibrocytes [78,323,324,325]. Highly intriguing in this context, low PTX-2 levels have been reported in patients with myelofibrosis [326].

Zinpentraxin alfa has shown anti-fibrotic and anti-inflammatory activity in preclinical models of fibrosing disorders [327,328,329], including prevention of bone marrow fibrosis by inhibiting bone-marrow-derived fibrocytic differentiation in vitro, reverting bone marrow fibrosis in animals, and also improving survival in a mouse model [329]. Importantly, zinpentraxin alfa has already demonstrated clinical activity and a well-tolerated safety profile in a phase-II trial in patients with pulmonary fibrosis [330,331,332]. Preliminary data from a safety and efficacy study of zinpentraxin alfa in patients with myelofibrosis are promising [308].

### 8.2. Colchicine

Circulating ECM markers may also be considered as indicators of treatment response in clinical trials on inexpensive old drugs, such as colchicine, statins, N-acetylcysteine (NAC) and metformin, which all have been shown to have the potential to reduce inflammation and organ fibrosis. Thus, colchicine is an established anti-inflammatory drug, which attenuates the NLRP3 inflammasome and also inhibits collagen synthesis [333,334,335]. The anti-inflammatory potential of colchicine has most recently been studied in patients with chronic coronary artery disease, displaying remarkable activity in terms of reducing cardiovascular events [335,336,337,338,339,340,341,342,343,344,345,346,347,348,349,350,351,352,353] and lowering a large number of inflammatory proteins, including CRP, IL1beta, and IL-6 [344]. Colchicine preferably accumulates in neutrophils, interfering with neutrophil adhesion, mobilization, and recruitment and inhibiting neutrophil chemotaxis and superoxide production. Accordingly, several circulating neutrophil granule components are strongly reduced during treatment with colchicine, including myeloblastin, carcinoembryonic antigen-related cell adhesion molecule 8, azurocidin, myeloperoxidase as well as circulating urokinase plasminogen activator surface receptor. Therefore, theoretically, colchicine may benefit patients with MPNs, which are characterized by the in vivo activation of circulating leukocytes and platelets, with a constant release of all the above neutrophil constituents, and hence, elevated circulating levels [2,13,15]. Of note, as previously alluded to, a significant association between plasma suPAR, serum PIIINP, serum hyaluronan and the serum concentration of the carboxyterminal telopeptide of type I collagen (S-ICTP) has been demonstrated, suggesting that enzymes of the uPA system participate in bone marrow remodelling in MPN [273]. YKL-40 (chitinase-3 like protein-1 (CHI3L1)-another biomarker of chronic inflammation, tissue remodelling, fibrosis and of disease burden in chronic inflammatory diseases and cancer, has also been demonstrated to be significantly elevated in patients with MPNs, the highest plasma levels being recorded in patients with myelofibrosis [302,303,304,305]. Whole-blood gene expression studies in MPNs have demonstrated a marked upregulation of several of the genes encoding the above proteins [354,355,356,357,358,359,360,361,362,363], including oxidative stress genes [360] and thromboinflammatory genes [364]. Amongst the latter is peptidyl arginine deiminase 4 (PAD4), which induces neutrophil extracellular trap formation (NETOsis) and protein citrullination [365]. In this regard, elevated circulating NETs have recently been reported in MPNs with significantly lower levels during treatment with IFN [366], which also suppresses upregulated thromboinflammatory genes [364], oxidative stress genes [361] and favourably impacts deregulated atherosclerosis and ECM genes in MPNs as well [361,362]. Importantly, colchicine also potently reduces NETosis activity [348] and oxidative stress [352]. Since NETs promote the differentiation and function of fibroblasts [367], the reduction in NETosis activity by colchicine may indirectly also impair fibrogenesis [348]. Other mechanisms for inhibition of fibrogenesis by colchicine include interference with type III procollagen synthesis [368] and increasing collagenase production [369]. Adding to this, colchicine suppresses the release of fibroblast growth factors from alveolar macrophages in vitro, including fibronectin and alveolar-macrophage-derived growth factor [370]. In the context of inducing regression of fibrosis in fibrosing disorders, such as idiopathic pulmonary fibrosis, clinical studies on the efficacy of colchicine have yielded discordant results, probably due to suboptimal study designs (e.g., retrospective studies; combination with other agents) but perhaps also because colchicine was initiated at an advanced stage of the fibrotic process [371,372,373]. To summarize, based on the above colchicine effects, we anticipate that colchicine should be conceived as a plausible modifier of not only the chronic inflammatory state and blood clot formation in MPNs, but may also favourably impact the resolution of bone marrow fibrosis.

### 8.3. N-Acetylcysteine (NAC)

The MPNs have been added to the list of “High Oxidative Stress“-diseases [23,359,374] since *JAK2V617F* mutation generates ROS [374,375] and also enhances oxidative stress via the impairment of mitochondrial antioxidative defence [376]. Taking into account that N-acetylcysteine (NAC) is a highly potent scavenger of ROS, NAC treatment has been proposed for the treatment of MPNs [374] not least also because NAC reduces thrombosis in a *JAK2V617F* knock-in mouse model and also reduces NET formation in neutrophils from patients with MPNs as well as controls [377]. Indeed, NAC may also favourably influence the deposition of collagen in the bone marrow, considering that NAC has shown efficacy in a subset of patients with idiopathic pulmonary fibrosis [378,379,380] although not being reproduced in a later study [381]. However, the beneficial potential of NAC in the treatment of idiopathic pulmonary fibrosis is still being pursued, the arguments and rationales being that a subset of patients with a particular genotype (TOLLIP rs3750920 TT), present in about 25% of patients with idiopathic pulmonary fibrosis, may actually gain benefits, since these patients seemed to benefit from NAC in a later subgroup analysis of the original cohort [382]. Intriguingly, an association between pulmonary fibrosis and myelofibrosis has been repeatedly described in the literature. Although this association may be explained via the seeding and sequestration of megakaryocytes and other hematopoietic progenitors in the pulmonary microcirculation [383,384,385] with ensuing release of growth factors mitogenic for fibroblast- and endothelial proliferation in the lungs [386], it is pertinent to consider, if some patients with MPN myelofibrosis and some patients with idiopathic pulmonary fibrosis may actually have some distinctive genotypes in common, which are associated with organ fibrosis. If so, the first step toward personalized medicine for patients with idiopathic pulmonary fibrosis may pave the path for similar initiatives in patients with myelofibrosis and associated neoplasms and also in autoimmune diseases like rheumatoid arthritis, in which pulmonary inflammation has, within recent years, been recognized as a long-term clinically silent feature [387,388,389,390]. In this scenario, linking genetics to measurements of ECM markers may turn out as a novel approach for personalized precision medicine, which hopefully may improve the selection of the right patients for the right anti-inflammatory and anti-fibrosing treatments. In addition, such combinations of diagnostic tools may also provide new insights into the disease mechanisms underlying pulmonary dysfunction in MPNs, considering that pulmonary mortality most recently has been shown to be a leading cause of death in patients with MPNs [391]. In this regard, it is worth speculating whether undiagnosed pulmonary fibrosis is a significant cause of pulmonary morbidity and mortality in a subset of patients. Concomitant studies of pulmonary function, imaging, inflammatory cytokines and circulating lung-derived ECM fragments are conceptually useful tools for studying pulmonary dysfunction and failure upon the clinical course and prognosis in MPNs.

### 8.4. Statins and Metformin

Similar to colchicine and NAC, statins possess potent anti-inflammatory and anti-thrombotic capabilities [392,393,394], and it has been argued that statins may impair clonal expansion and evolution in MPN [20,21,23,395,396,397,398,399], thereby potentially impeding disease progression and likely also the development of other cancers in MPNs [29,30,31,34,400]. Indeed, this assumption is supported by the Danish Statin Study on the impact of statins on cancer-associated mortality, which has shown a 15% reduction in cancer-associated mortality in statin users [401]. Furthermore, a most recent Danish register study has shown that statin treatment protects against the development of MPNs [402].

Several reviews in recent years have also underscored the potential of metformin as an anti-cancer agent [403,404,405,406,407]. In the context of tumor fibrosis as a barrier for effective tumor eradication by the immune system, it is highly interesting, that metformin seems to induce the depletion of collagen, thereby increasing the efficacy of chemotherapy in pancreatic cancer by degrading the tumor fibrosis barrier, which protects tumor cells from being attacked by effector immune cells [408].

For all the above reasons, clinical trials of the old, inexpensive drugs—colchicine, statins, NAC and metformin—in patients with newly diagnosed MPNs seem to be pertinent and timely. Such protocols should optimally include concurrent monitoring using conventional biomarkers of chronic inflammation (CRP, Neutrophil-Lymphocyte Ratio) [409], circulating levels of inflammatory cytokines [22], gene expression profiling studies [354,355,356,357,358,359,360,361,362,363,364], and ECM marker patterns [154] to describe in detail the associations between chronic inflammation, tissue remodelling and fibrosis development in MPNs. Of note, as addressed above, most recent studies have shown that both statins and metformin treatment protect against the development of MPNs, highlighting that these agents are potential disease-modifiers as well, likely due to suppression of the chronic inflammatory drive and ultimately, thereby also the development of bone marrow fibrosis [402,410]. Therefore, studies on circulating ECM protein fragments are of particular interest in individuals harboring the driver mutations, *JAK2V617F* and *CALR,* in the CHIP stage before the development of MPNs. In this regard, the Danish GESUS (General Suburban Study) cohort of 613 *JAK2V617F* and 35 *CALR*-positive CHIP-citizens are ideal cohorts [411,412] to study consecutively the dynamics of ECM proteins in a large, well-characterized cohort with a shortened life span compared to an age- and gender-matched CHIP-negative cohort. The decreased survival has been shown to be due to an increased risk of cardiovascular diseases and cancer (unpublished data). Measurements of circulating ECM fragments in this CHIP-cohort are expected to provide important information on the value of ECM fragments as predictors of cardiovascular diseases, development of MPNs, other cancers and other inflammation-mediated diseases, which are likely driven by somatic acquired inflammatory mutations, such as *JAK2V617F* and *CALR*. In such studies, the impact of various treatments (e.g., statins, bisphosphonates, NAC and metformin) should also be explored, assuming that a favourable treatment response translates into lower levels of, e.g., serum PIIINP in individuals being treated with these agents as compared to those not being treated. Our recent findings in the GESUS cohort, that statins reduce systemic markers of inflammation [395] and that individuals carrying the inflammatory and thrombogenic *JAK2V617F* mutation are exposed to increased oxidative stress and DNA damage [374,375,376], potentially implying an increased risk of cardiovascular diseases and development of not only MPN but also other cancers [413], emphasize the need for studies on new frontline disease markers to elucidate their value as biomarkers for early detection of among others cardiovascular diseases and cancer.

Future research on ECM protein metabolism in MPNs should focus on the value of circulating ECM fragments as tools to obtain an integrated signature of the impact of clonal myeloproliferation not only on bone marrow stroma changes in terms of fibrogenesis and angiogenesis but also systemically in other organs such as the heart, lungs and kidneys. However, it should be underscored that at this time, it is not possible to determine exactly the tissue origin of single markers or marker profiles, which is a key research issue to be resolved. It is hoped that the presently outlined studies, in conjunction with others, may contribute to defining the potential and future role of ECM markers in the management of the inflammatory and fibrosing processes that determine the transition from multiorgan dysfunction to multiorgan failure in MPNs [414]. As a class example, the ECM biomarker PRO-C6 (endotrophin) is a collagen VI formation-derived peptide, which most recently has been published as a potential next-generation biomarker in patients with heart failure with preserved ejection (HFpEF) [309]. This biomarker measures type VI collagen formation, which is upregulated, when fibroblasts are activated, giving rise to organ fibrosis. Endotrophin is a proinflammatory and profibrotic mediator, which is thought to contribute to HFpEF and tentatively also in MPN disease progression and development of myelofibrosis. Since HFpEF is prevalent among patients with chronic inflammatory diseases, such as hypertension and type II diabetes [310,311,312,313,314,315,316,317], it is intriguing to speculate if HFpEF is similarly prevalent in MPNs, and contributing to the increased cardiovascular disease burden. Concurrent echocardiographic studies and measurements of PRO-C6 are needed to explore the prevalence of HFpEF in MPNs and the potential of PRO-C6 as a novel biomarker for the assessment of individual patient risk and prognosis, but also as a novel tool in therapeutic decision making. In the context of the comorbidity–inflammation paradigm and MPNs as “A Human Inflammation Model “, MPNs and inflammation-mediated comorbidities may reinforce systemic inflammation and, accordingly the development of myocardial fibrosis and HFpEF [310,311,312,313,314,315,316]. This concept is supported by the progressive increase in plasma level of CRP with rising numbers of comorbidities in HFpEF patients. Similarly, recent studies have underscored the important role of chronic inflammation, assessed using serial measurements of CRP and NLR, for the development of MPNs with elevated levels of NLR several years before the diagnosis of MPNs and with a temporal increase in the number of comorbidities [409].

Neutrophil extracellular traps (NETs) have been shown to be induced by the potently inflammatory and thrombogenic *JAK2V617F* mutation [365,415], and most recently, a Danish study has reported that elevated NET levels decrease during treatment with pegylated interferon-alpha2 (IFN) [366]. Of note, NETs may promote fibroblast activation and fibrosis [367]. Considering that IFN decreases NETs [366], this effect may also facilitate a reduction in bone marrow fibrosis. Furthermore, IFN has been shown to alleviate angina pectoris in a patient with ischemic heart disease [416] and also favourably impacts deregulated atherosclerosis genes [362]. Accordingly, future research should engage in studies that concurrently investigate the deregulation of inflammation, NETosis, oxidative stress, antioxidative stress and atherosclerosis genes using whole-blood gene expression profiling in concert with serial measurements of circulating ECM biomarkers and levels of NETs in patients with MPNs and +/− HFpEF before and during treatment with IFN. Such studies will likely yield important new insights into common pathogenetic mechanisms in the development of myocardial fibrosis, its occurrence in patients with MPNs and the potential capability of IFN to revert myocardial fibrosis and bone marrow fibrosis.

As previously mentioned, incremental bone marrow fibrosis grade is today considered to herald a poor prognosis [78]. However, in some earlier studies, advanced bone marrow fibrosis per se did not appear to have a negative influence upon survival. On the contrary, among patients with classic PMF, those with diffuse collagenization of the bone marrow together with osteosclerosis actually fared better than those without osteomyelosclerosis [97]. This observation has not been pursued in other studies but may be of considerable interest since it may reflect that repair processes in the bone marrow in response to clonal, myeloproliferation may actually impair clonal expansion and thus postpone the acceleration of the disease [94,97]. Indeed, this early observation aligns with recent findings, supporting that both type III collagen and type VI collagen may have dual roles by being both pro- or anti-tumorigenic [417]. Thus, previous studies showed that a decrease in type III collagen in the tumor stroma may actually increase the aggressiveness of proliferative breast tumors [418], contrasting upregulated COL3A1 expression, which has been shown to associate with increased survival in breast cancer patients [419], suggesting that type III collagen can limit metastasis. Aligning these studies, most recently, tumor-derived type III collagen has been shown to be required in order to maintain tumor dormancy while disruption of type III collagen reactivates tumor cell proliferation [420]. Similarly, type VI collagen types VI-α1 and 2 chains can be anti-tumorigenic by inhibiting both proliferation, migration and invasion of bladder cancer cells [421]. An association between the attenuation of collagen synthesis in cancer-associated fibroblasts and an increase in tumor growth and spread has also been shown in several mouse studies [422,423,424,425]. Importantly, however, the resolution of tumor fibrosis is also considered to render the cancer cells more accessible to tumor killing by the immune system and targeted therapeutic intervention, including, e.g., with IFN or novel immunomodulating agents. Taken together, the progressive accrual of fibrotic tissue in the bone marrow in MPNs may actually not always be deleterious but rather exert a defence against clonal expansion and myeloproliferation, which, when eventually overwhelmed with the expansion and evolution of the malignant clone and ensuing accelerating myeloproliferation, is the ultimate outcome of balancing between the efforts, attempting to build up a barrier (fibrosis, osteosclerosis) against myeloproliferation and egress of immature progenitors from the bone marrow (metastasis) and resolution of fibrosis to facilitate the killing of malignant cells by the immune system. Thus, bone marrow fibroblasts and collagens in MPNs may have dual effects, which should be studied in further detail [70,72,73,154,417].

In 2023 and ahead, we have the advantage of exploring all these aspects more thoroughly via the concurrent measurement of circulating ECM fragments and studies of the bone marrow architecture, using a new technology, which much more precisely and comprehensively assesses bone marrow stroma changes, including fibrosis and angiogenesis, together with a description of cellular morphology with particular focus on the megakaryocyte lineage [426]. By using a machine learning approach, this novel technology allows much better quantification of reticulin fibrosis, thereby improving the early detection and monitoring of bone marrow fibrosis (Continuous Indexing of Fibrosis (CIF)) and thereby also improving MPN subtyping. Indeed, in combination with analysis of megakaryocyte morphology, CIF accurately discriminates between ET and pre-fibrotic myelofibrosis and also holds promise in the identification of MPN patients at risk of disease progression [427].

Concurrent studies of the associations between megakaryocyte morphology, CIF and ECM metabolites are expected to further refine disease classification and to improve the distinction between the MPN-subtypes [426,427,428,429,430,431,432,433], including the distinction between prefibrotic MF and genuine ET, further strengthening the evidence that prefibrotic MF is aligned along the biological MPN- continuum subtypes [434]. Furthermore, such studies are foreseen to provide highly important insights into the earliest changes in bone marrow stroma by studies of *JAK2V617F* and *CALR* positive individuals with CHIP. Thereby, we may also obtain novel information on the microenvironmental factors [435], which are influential for clonal expansion and evolution, both at the earliest stage in the CHIP phase and across the MPN disease spectrum from the early cancer phase to advanced myelofibrosis. This information on the determinant factors for remodeling of the bone marrow in the earliest stage of MPN development—the CHIP stage—may ultimatively be translated into novel therapies, targeting the chronic inflammatory state to prevent development of CHIP into MPNs and hopefully to prevent disease progression in the MPN-biological continuum as well [79]. In regard to monitoring the efficacy of novel agents, targeting not only the chronic inflammation, driving bone marrow fibrosis development, but also directly the ECM and the fibrotic process [436] CIF in combination with measurement of circulating ECM peptides, may be extremely important tools, for example in future studies of the role of inhibitors of lysyl oxidase (LOX), taking into account the LOX connection between megakaryocyte pathology and development of bone marrow fibrosis [437]. The importance of LOX was initially attributed to its role for the cross-linking of collagen or elastin and consequently in enhancing fibrosis. This cross-linking contributes to the accumulation of ECM by promoting intrapeptide and interpeptide chain crosslinking [437]. Importantly, LOX is also involved in proliferation of megakaryocytes and deposition of collagen fibers by oxidizing the PDGF receptor-β, amplifying its mitogenic potential [438] and accordingly stimulating megakaryocyte expansion by PDGF-BB. Therefore, LOX has been suggested as a new potential therapeutic target for myelofibrosis [78,439,440,441] based upon several experimental and clinical studies, not only in myelofibrosis and related myeloid neoplasms [439,442,443,444] but in other fibrosing diseases as well [445,446]. In addition to being first reported elevated in a mouse model of MPN [439], LOX has also been found elevated in patients with MPNs [447]. In MPN patients LOX levels may increase because of increased TGF-beta levels and signaling, since TGF-beta stimulates LOX expression [436,447,448]. Importantly, in addition to facilitating collagen crosslinking, LOX is also implicated in the development of thrombosis. Thus, elevated LOX augments platelet adhesion to collagen and platelet activation [436,449]. Accordingly, considering the heavy CVD burden in MPN and the potential of LOX to amplify receptor affinity for PDGF [439,441,450,451] LOX might also have a role in the pathogenesis and progression of CVD in MPN. Since inhibition of LOX may reduce bone marrow fibrosis and thrombosis in mice with experimental MPN and arterial thrombosis [436,439,449] clinical LOX-inhibitor trials are currently investigating or being planned with the issue to explore, whether LOX-inhibitors, either as monotherapy or in combination with, e.g., pegylated interferon-alpha2, JAK1-2 inhibitors or DNA-hypometylating agents may also reduce bone marrow fibrosis and thrombosis risk in patients with MPNs. In this regard concurrent studies on the associations between megakaryocyte morphology, CIF and ECM metabolites during treatment with LOX-inhibitors might unravel important novel aspects on the role of LOX and the impact of LOX-inhibition for development of bone marrow fibrosis in MPNs.

Theoretically, the optimal target for tumor killing is the cancer stem cell, which in myelofibrosis is the mutated bone marrow stem cell. As previously addressed, IFN-alpha2 is widely used in the treatment of MPNs and is able to induce minimal residual disease with normal bone marrow architecture in a subset of patients after long-term treatment (approximately 5 years) [240,241,244]. In concert with the normalization of elevated blood cell counts, a reduction in the potently inflammatory and thrombogen *JAK2V617F* mutation, lowering of myeloid-derived suppressor cells (MDSCs), boosting of immune cells, and downregulation of tumor antigens, IFN-alpha2 also favourably regulates several deregulated immune and inflammation genes, including the upregulation of HLA genes on tumor cells, thereby making them more accessible to killing by the immune system. In addition, IFN-alpha2 has also demonstrated a potent stem-cell-targeting potential by reactivating dormant *JAK2V617F*-positive stem cells and put them in cycle to be killed directly by IFN-alpha2 and IFN-activated immune cells (e.g., dendritic cells, NK-cells, T-cells) [240,241,244]. Interferon-beta (IFN-beta) has similar beneficial anti-cancer effects but is likely much stronger than IFN-alpha2, making it a potential alternative to IFN-alpha in the treatment of MPNs [452] and possibly also in the treatment of other cancers as well in combination with novel immunotherapies [453,454,455,456,457,458,459,460,461,462]. Remarkably, IFN-beta has been shown to suppress cancer stem cell properties in triple-negative breast cancer [460,461]. Combination therapy with IFN-alpha2 and the potent anti-inflammatory drug ruxolitinib (JAK1-2 inhibitor) has very strong synergistic effects with much faster normalization of peripheral blood cell count and a reduction in the *JAK2V617F* allelic burden in patients with MPNs [242,243,244]. Highly intriguing, normal bone marrow architecture with resolution of bone marrow fibrosis is achievable with this combination therapy, even within 12 or 24 months of therapy [242,243,244]. Similar combination therapies with cancer-stem cell targeting therapy (IFN-alpha2 or IFN-beta) together with a potent anti-inflammatory (e.g., JAK1-2 inhibitor, colchicine) and potentially tumor-fibrosis resolving therapy (e.g., colchicine, metformin, angiotensin II receptor blocker (losartan)) may favourably influence the impact of novel immunomodulating agents, such as immune-check inhibitors and inhibitors of TGF-beta signaling, in the treatment of cancer, e.g., pancreatic cancer and myelofibrosis (Table 2). In this context, studies of circulating ECM fragments before and during the above mono- or combination therapies are pertinent to study in-depth ECM protein metabolism and correlations to clinical, biochemical and molecular variables in order to discover novel treatment targets and novel predictive biomarkers for treatment response and prognosis. Such studies are also foreseen to add novel information on the role of the extracellular matrix and its composition for the efficacy of the above therapies, considering that collagen density has been shown to regulate the activity of tumor-infiltrating T cells [319] and modulates the immunosuppressive effects of macrophages [463]. These studies should also address the role of the various collagen fragments for T-cell suppression and T-cell exhaustion in myelofibrosis, considering that these fragments have been shown to mediate T-cell suppression in other cancers [464]. Table 3 summarizes key research questions to be addressed in future studies of circulating ECM fragments as biomarkers of clonal expansion and evolution in the CHIP-*JAK2V617F* stage and in the biological MPN disease spectrum from the early cancer stages, ET and PV, to the advanced myelofibrosis stage.
cancers-15-04323-t002_Table 2Table 2Commonalities and similarities between pancreas cancer (PC) and myelofibrosis (MF). Pancreas cancer and myelofibrosis are both strongly fibrosing neoplasms, characterized by dense fibrous tissue which surrounds and encapsulates the cancer cells. This fibrosis cap may have dual potentials in regard to tumor growth—both promoting tumor growth or inhibiting tumor growth—being dependent upon several factors, including the inflammatory state and collagen composition at a given time point. In both PC and MF, the fibrotic encapsulation may be a barrier for effective killing of cancer cells by the immune system. As such, PC and MF are unique model diseases for studying tumor stroma changes during cancer development. Already at the time of diagnosis PC is characterized by pronounced tumor fibrosis, which likely has developed during several months and even years in a subclinical precursor-cancer stage. In this context, MPNs are unique model diseases that have been described as “A Human Inflammation Model for Cancer Development “. Thus, advanced MF is preceded by decades of development from the early cancer stages (ET and PV) and even before ET with a long precursor-MPN stage as CHIP for decades as well. The inflammatory *JAK2V617F* mutation is prevalent in the CHIP stage (approximately 4% above + 50 years) and even more prevalent among high-risk patients for having MPNs (approximately 11% in stroke patients); concurrent studies of the dynamics of this somatic driver mutation and circulating ECM proteins in the CHIP-stage towards development of MPNs, associated comorbidities and other cancers and in the biological continuum from the early MPN-stages (ET, PV) towards the advanced “metastatic” MF stage are foreseen to yield unique novel insights into the dynamic evolution of inflammation-mediated organ dysfunction towards organ fibrosis, and these processes—chronic inflammation and fibrosis—being assessed via consecutive measurements of a broad range of ECM proteins. Commonalities and differences between the pancreas cancer fibrosis model and the MPN fibrosis model may provide novel information that might guide future studies in new drug targets and development of therapies to resolve tumor-related fibrosis and enhancement of the immune system to combat the cancer stem cell.
Pancreatic CancerMyelofibrosis**Stroma Composition**Type I, III and IV collagensEarly stage:Type III and IV collagenAdvanced stage:In addition Type I collagen**Stroma Changes**The normal basement membrane architecture is lost during PC progression and cancer cells are typically in direct contact with the interstitial matrix, i.e., type I and type III collagens.The normal basement membrane architecture is changed during progression of MPNs with the formation of continuous sheets below basement membranes; megakaryocytes are typically in direct contact with the interstitial matrix, i.e., type I and type III collagens and transmigrate together with CD34+ and progenitors the endothelial lining into the circulation to seed in the spleen, liver, lungs and elsewhere (“metastasis”).**Biomarkers of****Collagen Synthesis****Type III procollagen-peptides:** Elevated**S-PIIINP, S-Type IV, S-Laminin1:**Using ELISAs, levels of S-PIIINP, type IV antigen and laminin1 have been measured in MF and related neoplasms (please see text). These ELISAs measure type III collagen synthesis and synthesis of basement membrane constituents rather than degradation.**Biomarkers of****Collagen Degradation**C1M, C3M, C4M, C4M12a1: Elevated**C1M, C3M, C4M, C4M12a1:** to be performed**Therapies:**Stem-cell targeting; Immunomodulating; Anti-inflammatory Fibrosis-resolving.**Stem-cell targeting:** IFN-alpha2 or IFN-beta?**Immunomodulators:** Immune-check inhibitors, IFN-alpha2 or IFN-beta?; inhibitors of TGF-beta signalling **Anti-inflammatory agents** (e.g., Jak1-2 inhibitors)?**Fibrosis-resolving agents** (e.g., colchicine, metformin, angiotensin II blocker, statins, zinpentraxin alfa)?**Stem-cell targeting:** IFN-alpha2 or IFN-beta?**Immunomodulators:** Immune-check inhibitors, IFN-alpha2 or IFN-beta?; inhibitors of TGF-beta signalling.**Anti-inflammatory agents** (e.g., Jak1-2 inhibitors)?**Fibrosis-resolving agents** (e.g., colchicine, metformin, angiotensin II blocker, statins, zinpentraxin alfa)?**Abbreviation:** PC = Pancreatic cancer; myelofibrosis = MF; MPNs = myeloproliferative neoplasms; CHIP = clonal hematopoiesis of indeterminate potential; C1M, C3M, C4M, C4M12a1 = MMP-degraded fragments of collagen types 1, III, and IV, respectively.
cancers-15-04323-t003_Table 3Table 3Summary of key research questions to be addressed in future studies of circulating ECM peptides as biomarkers of clonal expansion and evolution in the CHIP-*JAK2V617F* stage and in the biological MPN-continuum from the early cancer stages, ET and PV, to the advanced myelofibrosis stage.Research QuestionsDoes measurement of circulating ECM peptides have the potential to differentiate genuine ET from prefibrotic myelofibrosis, prefibrotic myelofibrosis from classic myelofibrosis and predict leukemic transformation in MPNs?Do distinct circulating ECM peptides or ECM peptide signatures depict disease activity/burden as measured by conventional parameters (leukocyte and platelet counts, *JAK2V617F* allelic burden, plasma lactic dehydrogenase level)Do circulating ECM peptides or ECM peptide signatures associate with antecedent thrombosis and/or predict thrombosis riskHow does artificial intelligence-based morphological fingerprinting of megakaryocytes and “Continuous Indexing of Fibrosis” correlate with circulating ECM peptide signatures?Do circulating ECM peptides reflect comorbidity burden in MPNs?Do circulating ECM peptides or ECM peptide signatures predict treatment response (hydroxyurea, IFN-alpha2, JAK1-2 inhibitor (ruxolitinib or fedratanib), statin (atorvastatin), combination therapy (IFN-alpha2 + ruxolitinib), epitope -vaccination)?Do CHIP-*JAK2V617F* positive citizens have a distinct ECM peptide signature as compared to CHIP-*JAK2V617F* negative citizens?Does measurement of circulating ECM peptides in the CHIP stage have the potential to diagnosis early development of MPNs?Does measurement of circulating ECM peptides in the CHIP stage have the potential to diagnosis early development of inflammation-mediated comorbidities and associated organ dysfunction and incipient organ fibrosis in diseases such as dementia, age-related macular degeneration, aneurysms, myocardial fibrosis with heart failure (HFpEF), pulmonary dysfunction/failure, chronic kidney failure or osteopenia?Does measurement of circulating ECM peptides in the CHIP stage have the potential of early detection of development of MPNs or another cancer?Do circulating ECM peptides or ECM peptide signatures associate with distinct immune cell profiles, including a T-cell exhaustion signature?Do circulating ECM peptides or ECM peptide signatures predict overall survival in the CHIP stage and in MPNs?**Abbreviations**: HFpEF: heart failure with preserved ejection fraction, which is prevalent in patients with chronic inflammatory diseases, and likely, therefore, in MPNs as well, considering that MPNs are “inflammatory” neoplasms with a huge inflammation-mediated comorbidity burden.

## 9. Conclusions

MPNs are associated with an evolving chronic inflammatory state, concurrent immunoderegulation and fibrosis [20,21,22,23,24,25,26,27,28], which are often associated with gradually emerging chronic inflammation-related comorbidities [16,17,18,20,21,23,29,30,31,32,33,34,35,37,38,39,40,41,42,43,44,45,46,47,48,49,50,51,52,53,54,55,56,57,58,59,60,61,62,63,64]. Within the last decade, novel sensitive and reproducible assay technologies for quantification of circulating ECM metabolites have been introduced for early detection and follow-up in fibrosing conditions, e.g., myelofibrosis and allied conditions including cancer-related desmoplasia. At the same time, there have been considerable gains in the knowledge about pathogenetic pathways in the MPNs, and their natural history has been elucidated in more detail and classification criteria have been refined. It therefore seems pertinent to take advantage of the current state-of-the-art clinical and laboratory opportunities by resuming previous research efforts to identify candidate ECM markers or marker patterns, aiming at early detection and prognostication of MPN and related comorbidities in large, clinically well-characterized study populations. Combined with innovative novel advanced artificial intelligence approaches, demonstrating the potential of extracting quantitative data from routinely prepared bone marrow biopsies much more reliably and precisely than conventional histopathology, thereby improving the assessment and classification of MPN patients [426,427], such studies are foreseen to offer new insights into the pathobiology of MPNs and how to interrupt the vicious, inflammatory self-perpetuating circle, that otherwise will evolve into bone marrow failure and development of inflammation-mediated multimorbidities [16,17,18,20,21,23,29,30,31,32,33,34,35,37,38,39,40,41,42,43,44,45,46,47,48,49,50,51,52,53,54,55,56,57,58,59,60,61,62,63,64]. In the context of MPNs sharing with type II diabetes mellitus (DM) so many inflammation-mediated comorbidities, and both diseases being heavily burdened by multimorbidities and multiorgan failure in the advanced disease phases [71], endotrophin may be one of the promising markers to investigate in MPNs, considering its excellent performance as a marker of complications in type 2 DM [465].

## Data Availability

The data presented in this study are available in this article.

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
