# Peer review of "Revisiting Circulating Extracellular Matrix Fragments as Disease Markers in Myelofibrosis and Related Neoplasms"

_cancers, 2023, doi:10.3390/cancers15174323_

Round 1
Reviewer 1 Report
This is an excellent piece of work, with a comprehensive and systematic review of the importance of the extracellular matrix (ECM) in MPN and the critcal role of inflammation and stromal remodelling in these diseases. The authors make a compelling case that a better understaning of ECM biology can lead to improved biomarkers of MPN diagnosis /disease progression and inform the search for improved therapies. The article is exhaustively and appropriately referenced and clearly draws upon the experience and expertise of the authors. I have no hesitation in recommending this article for acceptance.
The English is generally very good with only minor editing required.
Author Response
We are most grateful for this Reviewer’s comments, that our manuscript is excellent and provides a comprehensive and systematic review of the importance of the extracellular matrix (ECM) in MPNs. We are happy that this Reviewer recommends our paper for publication. In regard to the English language this Reviewer notes that it is generally very good with only minor editing required. We have made minor corrections in the revised manuscript.
Reviewer 2 Report
In this extensive and well researched review by Hasselbalch et al, the authors describe in detail the concept of circulating extracellular matrix fragments as a biomarker of myelofibrosis disease activity. The review is exhaustive in its detail of connective tissue response to injury and studies on extracellular matrix metabolism in MPNs. What is most welcomed, however, is the discussion on rationale for revisiting ECM biomarkers. As someone interested in this area, I learned quite a deal about this topic from this monograph and congratulate the authors on a well-researched and thoughtful review. I have no comments and welcome this addition to the literature
Author Response
Thanks so much for this Reviewer’s very positive comments . We are happy that this Reviewer finds our Review exhaustive, well researched and in particular welcomes our discussion on rationale for revisiting ECM biomarkers. We note with pleasure that this Reviewer - with interest in the area - has learned from reading our paper, congratulates us on a well-researched and thoughtful review with no comments but welcomes our work to the literature on ECM biomarkers in MPNs.
Reviewer 3 Report
Dr. Hasselbalch is a clinician scientist internationally recognized for his numerous and high-quality contributions to the field of myeloproliferative disorders. In the present review, Dr. Hasselbalch and colleagues review current knowledge on the use of the levels of extracellular matrix fragments in the circulation as biomarkers for disease progression and response to therapy in patients with myelofibrosis. Concepts are discussed side by side with results with cancers of solid organs which present, as myelofibrosis, increased deposition of extracellular matrix components in the tumors.
The review is accurate, and the literature discussed is comprehensive. The structure of the review is robust and provides a balanced overview of the field. Below are some specific comments.
Introduction, second paragraph: It is highly recommended to tune down the statement: “MPNs arise from clonal hematopoiesis of indeterminate potential (CHIP), which is an inevitable consequence of normal aging and is defined as the presence of a clonal mutation in a driver gene, occurring with a variant burden of ≥ 2% but without any clinical evidence of a hematologic cancer”. This sentence is supported only by a review. The role of CHIP in driving the generation of immune-biased hematopoietic stem/progenitor cells that are responsible for the hematological defects of the elderly and that may contribute to the insurgence of hematologic cancers is presently unclear and is an area of active investigation. The genetic, and epigenetic, mechanisms that determine the establishment of CHIP with age are far from being defined. Since the mechanisms which determine CHIP are still controversial and CHIP is outside the areas directly covered by the review, this paragraph should be either deleted or reworded in more general terms.
Introduction, third paragraph: The discussion of “inflammation” as driver of MPN is in general well organized. The comments I have for this session are: 1) reword for precision the following statement: “Increased risk of fractures, likely mediated by the chronic inflammatory state”. As discussed in refs 50, inflammation increases bone deposition, activating osteoblast proliferation and the deposition of extracellular matrix components while impairing calcium metabolism and deposition. As result, the bones are larger by more fragile. This is implicitly mentioned later on when it is stated that the patients are “osteopenic” (reduced calcium levels). Whether osteopenia is a direct consequence of MPN is however controversial because age and sex are confounding factors in discussing the cause of osteopenia in old individuals. On the other hand, discussion of bone in MPN patients is relevant for this review because the increased deposition of the proteins of the extracellular matrix observed in the bone of MPN may contribute, together with fibrosis, to increase the extracellular matrix fragments present in the circulation. 2) the discussion of the role of proinflammatory cytokines in MPN should include interleukin 8.
Introduction, fourth paragraph: the opening sentence in incomplete. Myelofibrosis does not arise only from ET, as stated, but also from Polycythemia Vera and from an indolent form of myelofibrosis defined pre-MF. In addition, fibrosis is not only observed in the bone marrow but also in the spleen and other sites of extramedullary hematopoiesis. This is relevant because, by contrast with the bone marrow in which fibrosis is mostly associated with reticulinic fibers, in the extramedullary sites fibrosis is often due to collagen fibers.
Figure 1: The role of lysil oxidase produced by megakaryocytes in the formation of collagen fibers is poorly discussed. The review by Dr. Ravid on this subject should be discussed (Blood. 2012 Aug 30; 120(9): 1774–1781). This is important because of the clinical trials which have been conducted with inhibitors of Lysil Oxidase as antifibrotic agents in myelofibrosis. In addition, the progression from reticulin fibers to collagen should be indicated more clearly. it is not
Lines 230-254. This paragraph is convoluted and repetitive. It could be written more concisely.
Figure 2: Vessels are not indicated and their structure appears similar in Healthy and Myelofibrosis bone marrow. Megakaryocytes are not present in the tumor fibrosis. This is a big difference. They are present in the fibrotic spleen of myelofibrosis patients. What are the cells present in the tumor fibrosis and not in the healthy solid tissue?
Lines 276-283. It is not clear how circulating ECM fragments may be useful to discriminate ET from pre-myelofibrosis, since both conditions are not associated with fiber deposition. The alternative approach that the megakaryocyte morphology may represent a possible marker to discriminate between ET and pre-myelofibrosis should also be discussed (Barosi and Balduini’s studies).
Table I. Although the spirit beyond Table I is greatly appreciated, in practice its design is found disappointing. It is a list of individual studies, some of which have analyzes only one end point, while others have analyzed more than one end-point, It should instead be a rational organization of results obtained for each end-point in multiple studies. Since the number of patients analyzed in the individual studies is in general not robust but most of the results are concordant, the presentation of the data as a summary of individual studies would make the conclusion much stronger. It is suggested to organize the table one end point per line (S-PIIINP, S-Laminin 1, S-PICP, S-TIVC, S-HYA, U-HYPRL, etc) and 4 column., the second column should present the cumulative numbers of patients analyzed by different studies, the third column should summarize the results, the fourth column should provide Comments/Conclusions, the fifth column all the refs on that particular end-point. In the case of contradictory results, the lines for a specific end-point could be increased to discuss positive, neutral and negative correlations.
Table 2. This table is confusing and overall weak because only one end point (Type III collagen pro-peptides) has been analyzed both in myelofibrosis and in some of the solid cancers discussed (Breat, Colon, Pancreas, Liver, Melanoma). This end-point has not been studied in gastric and ovarian tumors and in SCLC and NSCLC. The other pro-peptides studied in solid cancers described in this table have not been studies in MPNs. For this reason, it is strongly suggested to delete this table and to eventually summarize its content in the text.
Table 3. This table is a nice complement to Figure 2 which also compares fibrosis in MPNs and in pancreatic cancer. It appears that the most extensive comparison between the features of fibrosis between myelofibrosis and solid cancers was done for pancreatic cancer (Figure 2 and this Table). I strongly recommend to focus the review only on the comparison between myelofibrosis and pancreatic cancer.
Table 4 (Research Questions). Research questions are welcome. However, they should be prioritized in the order of relevance and feasibility. Discussion of feasibility may require provisional assessment of the power (number of patients) necessary to achieve robust conclusions for each question. Not all of these questions may be currently feasible.
Minor Comments
Table 1: The number and type of patients reported in column 2 is not consistently reported across studies and does not always reflect the results described in the column. In the case of Refs: 278: it is not clear what are the observations reported related to S-PIIIINP. It is highly recommended to check the information for additional inconsistencies.
Round 2
Reviewer 3 Report
Thanks, Overall, my comments were rationally addressed.